# Is increasing ice crystal sedimentation velocity in geoengineering simulations a good proxy for cirrus cloud seeding?

Blaž Gasparini[1], Steffen Münch[1], Laure Poncet[1], Monika Feldmann[1], and Ulrike Lohmann[1]

[1]Institute for Atmospheric and Climate Science, ETH Zürich

*Correspondence to:* Blaž Gasparini (blaz.gasparini@env.ethz.ch)

**Abstract.** The complex microphysical details of cirrus seeding with ice nucleating particles (INPs) in numerical simulations are often mimicked by increasing ice crystal sedimentation velocities. So far it has not been tested whether these results are comparable to geoengineering simulations in which cirrus clouds are seeded with INPs. We compare simulations where the ice crystal sedimentation velocity is increased at temperatures colder than -35°C with simulations of cirrus seeding with INPs using the ECHAM-HAM general circulation model. The radiative flux response of the two methods shows a similar behaviour in terms of annual and seasonal averages. Both methods decrease surface temperature but increase precipitation in response to a decreased atmospheric stability. Moreover, simulations of seeding with INPs lead to a decrease in liquid clouds, which counteracts part of the cooling due to changes in cirrus clouds. The liquid cloud response is largely avoided in a simulation where seeding occurs during night only. Simulations with increased ice crystal sedimentation velocity, on the contrary, lead to counteracting mixed-phase cloud responses. The increased sedimentation velocity simulations can counteract up to 60% of the radiative effect of $CO_2$ doubling with a maximum net top-of-the-atmosphere forcing of -2.2 W m$^{-2}$. They induce a 30% larger surface temperature response, due to their lower altitude of maximum diabatic forcing compared with simulations of seeding with INPs.

# 1 Introduction

Cirrus seeding is a proposed geoengineering method to decrease the occurrence of cirrus clouds by changing their optical properties. Cirrus clouds on average have a stronger longwave (LW) than shortwave (SW) effect on the radiative balance, leading to a positive net cloud radiative effect (CRE), as estimated from satellite data (Hartmann et al., 1992; Chen et al., 2000; Futyan et al., 2005; Hong et al., 2016; Matus and L'Ecuyer, 2017), in situ lidar observations (Kienast-Sjögren et al., 2016), and global modelling studies (Gasparini and Lohmann, 2016). Thus a reduced amount of cirrus clouds will increase the amount of outgoing longwave (LW) radiation (Mitchell and Finnegan, 2009) and thereby cool the climate. Cirrus CRE has a pronounced seasonal and daily cycle, with higher values in the winter hemisphere (or at night) where the reflection of SW radiation is limited by the lack of insolation. We define cirrus clouds as all clouds that form at temperatures lower than -35°C with no additional altitude criteria.

Two microphysical formation pathways of cirrus clouds exist:

– Homogeneous freezing of solution droplets occurs at high relative humidities with respect to ice ($RH_{ice}$) and can lead to a large number of ice crystals (ICs) depending on temperature and updraft velocity (Kärcher and Lohmann, 2002). If their concentration is large, their growth is limited, as they rapidly consume the available water vapour (Ickes et al., 2015).

– Heterogeneous freezing can occur in the presence of effective INPs which lowers the freezing energy barrier, allowing droplets to freeze at lower $RH_{ice}$ and/or smaller updraft velocities (Kärcher and Ström, 2003; Hoose and Möhler, 2012).

Heterogeneous ice nucleation can suppress homogeneous nucleation in conditions of slow updrafts, commonly found in the upper troposphere (Jensen et al., 2016; Kärcher and Ström, 2003), resulting in optically thinner and shorter-lived cirrus clouds. A modelling study by Lohmann et al. (2008) showed that the global net top-of-the-atmosphere (TOA) radiative balance can change by up to 2.8 W m$^{-2}$ as a result of a complete shift from homogeneous to heterogeneous cirrus formation (the numbers are reported in Mitchell and Finnegan (2009)). As the upper tropospheric INP number concentrations is limited (DeMott et al., 2003), only few ICs can nucleate heterogeneously and consequently grow to larger sizes (Kuebbeler et al., 2014).

A large fraction of cirrus clouds at temperatures warmer than -60°C is found to have formed from homogeneous nucleation of cloud droplets in convective clouds forming anvil cirrus (Jensen et al., 2015). These cannot be modified by seeding of INPs as their formation is dominated by strong updraft velocities (Penner et al., 2015). In addition, ICs in warm cirrus in the extratropics often form by heterogeneous freezing of cloud droplets in mixed-phase clouds, which are subsequently advected to cirrus conditions (Luebke et al., 2016; Wernli et al., 2016; Voigt et al., 2016). Cirrus seeding can perturb only the nucleation of ice crystals in supersaturated cloud-free conditions (in-situ formed cirrus) and their subsequent initial growth.

Cirrus seeding tries to modify the competition between homogeneously and heterogeneously formed ICs by artificial injections of efficient INPs with the goal of cooling the climate. Modelling studies by Storelvmo and Herger (2014) and Storelvmo et al. (2014) suggested that cirrus seeding can decrease the net TOA radiative balance by up to $2\,\mathrm{W\,m^{-2}}$ or decrease the surface temperature by up to $1.4\,^\circ\mathrm{C}$. On the other hand, a study by Penner et al. (2015) showed no significant net radiative change as a result of seeding due to a larger concentration of upper tropospheric INPs in their reference climate, no upper limit on the subgrid-scale updraft velocities, and the inclusion of the competition of pre-existing ICs for the available water vapour. Gasparini and Lohmann (2016) also found an insignificant radiative response to cirrus seeding in their simulations. They attributed it to a decrease in IC radius and an increase in cirrus cloud cover by forming new cirrus in previously cloud-free ice supersaturated regions.

As it is computationally demanding to simulate the detailed cirrus microphysical processes, climatic responses of seeding are often represented by increasing the IC sedimentation velocity in cirrus clouds (Muri et al., 2014; Crook et al., 2015; Jackson et al., 2016). Increasing the IC sedimentation velocity can, analogous to seeding, decrease the amount of cirrus cloud cover, ice water content (IWC), and ice crystal number concentration (ICNC). Such a modelling strategy was also selected by the Geoengineering Modeling Intercomparison Project (Kravitz et al., 2015). However, it has never been systematically analysed whether this method leads to results comparable to seeding with INPs.

In this paper we compare the radiative, microphysical, and climatic responses between the increased sedimentation velocity and seeding simulations with the help of suitable INPs. We point out differences between the two setups, examine liquid and mixed-phase cloud responses to changes in cirrus clouds, and show geographical areas where both ways of simulating cirrus geoengineering are most effective. We also evaluate the maximum effect of the increased sedimentation velocity schemes.

## 2  Methods

### 2.1  Model setup

We use the ECHAM6-HAM2 aerosol-climate model (Stevens et al., 2013; Zhang et al., 2012; Neubauer et al., 2014) with a horizontal resolution of $1.875^\circ \times 1.875^\circ$, 31 vertical levels, and a model timestep of 6 minutes. The model top is at 10 hPa. The level thickness at typical cirrus altitudes varies between 500 and 1000 m. The two-moment aerosol scheme (Vignati et al., 2004; Stier et al., 2005) interactively simulates aerosol emissions, their growth, coagulation and sink processes in terms of their number and mass mixing ratios. The model uses a two-moment cloud microphysics scheme with prognostic equations for cloud liquid and ice mass mixing ratios as well as cloud droplet and ice crystal number concentrations (Lohmann et al., 2007). The cirrus nucleation scheme by Kärcher et al. (2006) simulates the competition between homogeneous freezing, heterogeneous freezing, and deposition of water vapour on pre-existing ICs. Heterogeneous freezing occurs via deposition nucleation of insoluble coarse and accumulation mode dust aerosols or immersion freezing of internally mixed (coated) dust aerosols based on laboratory measurements by Möhler et al. (2006, 2008). The formulation of vertical velocity used for cirrus cloud formation

considers the large-scale velocity field and a subgrid-scale contribution derived from the turbulent kinetic energy. The latter is replaced by a gravity wave parametrization by Joos et al. (2008) over mountain regions. The detailed implementation of the cirrus formation scheme in the ECHAM-HAM general circulation model has been described in Kuebbeler et al. (2014) and Gasparini and Lohmann (2016).

We use the convective mass flux scheme of Tiedtke (1989) with modifications for deep convection from Nordeng (1994), which is an important source of detrained cloud ice leading to frequent anvil cirrus formation. The model gridboxes are considered partially cloudy above a certain relative humidity threshold, and fully cloud covered when relative humidity reaches 100%, following Sundqvist et al. (1989).

## 2.2  Experimental setup

For the idealized seeding scenario we perform simulations where the sedimentation velocity of all ICs at temperatures below -35°C is increased by factors of 2 (simulation VEL2), 4 (VEL4), and 8 (VEL8), and two simulations where the sedimentation velocity is either always set to 2 m s$^{-1}$ (VELmax) or only during night (VELmaxN). 2 m s$^{-1}$ is the maximum sedimentation velocity ICs can achieve in our model. The sedimentation velocity increase applies for all the cirrus ICs, regardless of their microphysical origin. We always show anomalies with respect to the unperturbed reference simulation (REF). Fig. 1 shows the
relation of IC size and their sedimentation velocity following the formulation by Spichtinger and Gierens (2009) for typical upper tropospheric conditions in the tropics. The upper sedimentation velocity limit of 2 m s$^{-1}$ used in ECHAM-HAM does not significantly influence our results, as ICs with radii smaller than 90 $\mu$m in atmospherically relevant conditions do not sediment faster than approximately 0.3 m s$^{-1}$. Only an IC of about 1 mm radius would fall with a velocity of about 2 m s$^{-1}$. Moreover, ice crystals larger than 90 $\mu$m are transferred from ice into snow (Levkov et al., 1992) and precipitate out of the atmosphere
within one model timestep.

For the realistic seeding scenarios, we performed 5 simulations of globally uniform continuous seeding in areas with temperatures colder than -35°C as decribed by Gasparini and Lohmann (2016). In our simulations we either increase the cirrus IC sedimentation velocity or seed with geoengineered INPs, which sediment with the size dependent sedimentation velocities (Spichtinger and Gierens, 2009). In SEED simulations we use INPs with the modal radius of 0.5 $\mu$m, while in SEEDr50
simulations the radius is increased to 50 $\mu$m. In this way we overcame the cloud cover increase and IC radius decrease that we see when seeding with 0.5 $\mu$m particles (Gasparini and Lohmann, 2016). We do not go beyond a radius of 50 $\mu$m despite using even larger INP sizes would likely result in larger climatic impacts. As the injected particle mass increases cubically with particle size, its practical use will be limited due to the needed delivery into the upper troposphere and shorter atmospheric
residence time. We seed all areas supersaturated with respect to ice at temperatures below -35°C with 0.1, 0.3, 1, 3, 10, 30, and 100 INPs L$^{-1}$ (consequently we name the simulations as SEED0.1r50, SEED0.3r50, SEED1r50, etc., see Table 1), which nucleate in deposition mode at RH$_{ice}$ as low as 105% (Mitchell and Finnegan, 2009). In addition, we simulate a scenario where 50 $\mu$m INP of 1 INP L$^{-1}$ is applied only during night (SEED1r50N). It is important to note that we only modify in situ formed cirrus and not the convective anvil clouds as in situ deposition nucleation does not occur in anvils. Furthermore, the injected

INPs do not interact with radiation and cannot directly influence mixed-phase or liquid clouds.

In addition to the standard model radiative fluxes, we separately diagnosed the cloud radiative effect contribution of clouds at temperatures colder than -35°C (cirrus cloud radiative effect, cCRE) with the help of the double call of the radiation routine. Similarly, we diagnosed mixed-phase cloud radiative effects (mpCRE) for all clouds at temperatures between -35°C and 0°C independent of their cloud phase, and liquid cloud radiative effects (liqCRE) for clouds at temperatures above the freezing level.

All simulations are after a 3-month model spin-up run for 5 years with fixed sea surface temperatures (SST) to study radiative flux anomalies and fast responses to seeding. Simulations SEED1r50 and VEL2 are extended to 10 years to increase the statistical robustness of the results. They are additionally simulated in the mixed layer ocean (MLO) setup in order to study long term microphysical and climatic responses, especially temperture and precipitation. The MLO simulations are run for 50 years, but we only assess the anomalies of the last 30 simulated years, after the model has reached an equilibrium. A list of all simulations and their specifications can be found in the Table 1. The significance is calculated based on a double-sided Welch's t-test at the 95% significance level.

# 3 Results

## 3.1 Cirrus geoengineering

### 3.1.1 Increased sedimentation velocity

The radiative effects decrease exponentially with the increase in sedimentation velocity (Fig. 2a) as already noted by Jackson et al. (2016). This is because the cirrus CRE always decreases by about 30% when doubling the IC sedimentation velocity, i.e. comparing REF with VEL2, VEL2 with VEL4, or VEL4 with VEL8 (Table 2). In the VELmax simulation increasing IC sedimentation velocities in cirrus to 2 m s$^{-1}$ leads to a negative net TOA radiative balance anomaly of -2.20 W m$^{-2}$ $\pm$ 0.26 W m$^{-2}$ which corresponds to about 60% of the radiative forcing induced by the doubling of the $CO_2$ concentrations (Stocker, 2013).

In VELmaxN we only increase the IC sedimentation velocity in cirrus during night, which leads to an overall (considering day and night) 15-20% smaller radiative effect decrease compared with VELmax . The result is consistent with the cirrus CRE diurnal cycle diagnosed from the model, which reaches 8 W m$^{-2}$ in the global annual average at night and 1 W m$^{-2}$ during day, when cirrus reflect part of the incoming SW radiation (Fig. 3). The VELmax simulation, which sets the cirrus IC sedimentation velocity to the unrealistically high value (Fig. 1), shows that globally uniform cirrus cloud thinning can reduce the cirrus CRE by about 3.3 W m$^{-2}$ which is equivalent to ∼75% of its full value (Table 3).

### 3.1.2 Cirrus seeding with ice nucleating particles

In the SEED simulations we inject seeding INPs of 0.5 $\mu$m radius at every model timestep in areas with temperatures colder than -35°C. We see no significant radiative response for concentrations of up to 1 INP L$^{-1}$ (Fig. 2b), while a net TOA positive radiative anomaly develops by seeding with more than 3 INPs L$^{-1}$ (overseeding) as explained in detail in Gasparini and Lohmann (2016). With an increased radius of 20 and 50 $\mu$m (simulations SEEDr20 and SEEDr50) and a seeding concentration of 1 INP L$^{-1}$, we achieve a significant negative TOA radiative anomaly of -0.46 $\pm$ 0.14 W m$^{-2}$ and -0.85 $\pm$ 0.40 W m$^{-2}$, respectively (Fig. 2c and Table 4). Seeding with large INPs leads to larger newly formed heterogeneous ICs and therefore avoids their decrease in size as observed in Gasparini and Lohmann (2016). Moreover, the initial increase in cloud cover by seeding of ice supersaturated clear sky regions with efficient INPs is outweighted by the large increase in the IC sedimentation velocities, which leads to a net cirrus cloud cover decrease (Fig. 5a).

Large INPs have a shorter atmospheric lifetime because of the quadratic dependence of particle fallspeed on its radius where the impact of turbulence can be neglected as we are in Stokes regime. For instance, the vertical velocity of a 0.5 $\mu$m aerosol particle (considering a density of 2500 kg m$^{-3}$, similar to dust aerosols) in the upper troposphere is $\sim$10$^{-4}$ m s$^{-1}$, while a 50 $\mu$m particle falls with a velocity of 1 m s$^{-1}$. On the other hand, the probability of one INP to freeze in a given time as described by the classical nucleation theory depends on the INP's surface area which increases quadratically with particle size. Quadratic fallspeed velocity and freezing probability increases cancel out each other leading to no change in the concentration of ICs formed on geoengineered INPs when they increase in size.

The large size of seeding INPs also increases the deposition flux of water vapour onto the INP leading to a more effective drying of the upper atmosphere. The largest disadvantage of seeding with large INPs is the cubic dependence of particle mass on its radius: an increase in radius from 0.5 to 50 $\mu$m increases its mass by a factor of $10^6$, making the transport of the seeding material to the upper troposphere much more challenging. Additionally, the seeding frequency of the large INPs would probably need to be larger compared with the small INPs seeding, due to their faster sedimentation.

The radiative anomalies obtained by injecting 0.3 or 3 INPs L$^{-1}$ of 50 $\mu$m radius are -0.66 $\pm$ 0.35 W m$^{-2}$ and -0.77 $\pm$ 0.27 W m$^{-2}$ and thus are not significantly different from those with injecting 1 INP L$^{-1}$ (Fig. 2c). The effective seeding range with similar results thus spans over about an order of magnitude of INP concentrations. The response from a simulation with large particle seeding only at night (SEED1r50N) with a net TOA radiative anomaly of -0.91 W m$^{-2}$ is similar to the one from SEED1r50.

Interestingly, our cirrus clouds respond differently compared to what has been described in Storelvmo and Herger (2014). A solar zenith angle dependent seeding scenario which seeds about 40% of the earth's surface leads, unlike the cited study, to only about half of the radiative flux anomaly compared with the globally uniform seeding scenario SEED1r50. The difference probably originates from the radiative effects of the cirrus clouds in ECHAM-HAM, which show a peak over the tropics (Fig.

3) and, differently from Storelvmo and Herger (2014), a large proportion of tropical cirrus clouds is formed by homogeneous freezing (Gasparini and Lohmann, 2016).

## 3.2 Response comparison

### 3.2.1 Radiative and microphysical responses

We now focus on the climatic and microphysical responses of the seeding with 1 INP L$^{-1}$ (SEED1r50) and increased sedimentation velocity (VEL2) scenarios due to their similar TOA net radiative flux anomalies ($\sim$ -0.8 W m$^{-2}$, Fig. 4a) in 10-year fixed SST simulations. Both geoengineering simulations show a positive anomaly in net SW TOA fluxes, due to a smaller SW CRE, i.e. less SW radiation reflected by cirrus clouds. The LW radiation budget, on the other hand, is more negative due to the increased outgoing LW radiation in response to a decrease in cirrus cloud cover. The radiative changes result in increased tropospheric cooling, decreasing the atmospheric stability, increasing convection, and leading to a precipitation increase of about 1% for both VEL2 and SEED1r50 (Fig. 4b).

Interestingly, the simulation SEED1r50N leads to a slightly larger net TOA radiative anomaly, which is different from the comparison of VELmaxN and VELmax simulations and counterintuitive as the cirrus cloud radiative effects (cCRE) in ECHAM-HAM on average is positive also during the day. However, while the impact of increasing the sedimentation velocity only during night ceases immediately when the sun appears above the horizon, the seeding of INPs has some inertia. The effective cirrus cloud lifetime is in the range of several hours as diagnosed from our model, with values around 6 hours in the tropical tropopause region. This is shorter compared with available studies which estimated it to 12-30 hours (Luo and Rossow, 2004; Jensen et al., 2011; Gehlot and Quaas, 2012). We therefore expect the seeded clouds to prevent the formation of homogeneous cirrus also some hours after sunrise, when the sun is low on the horizon and the cirrus LW warming effect significantly outweighs the cooling by the scattering of the SW radiation. Moreover, the simulation SEED1r50N avoids the warming effect induced by a response of liquid clouds to seeding during the day, as described in the Sect. 3.2.2.

From now on we focus only on the 30-year MLO simulation anomalies. The surface temperature decrease in response to changes in the atmosphere results in a more stable lower troposphere compared with fixed SST simulations. This overcompensates the fast (surface temperature independent) precipitation response in Fig. 4b, leading to a small decrease in global average precipitation for both scenarios (Fig. 4d).

Cirrus cloud cover decreases in both simulations (Fig. 5 a,b) with maximum anomalies between 7 (extratropics) and 15 km (tropics) altitude. The maximum decrease occurs at about 11 km altitude amounting to 4% in SEED1r50, whereas the decrease is about 3-4 times smaller in the VEL2 scenario (Fig. 5e). Both simulations show an IWC anomaly of about -0.2 to -0.5 mg kg$^{-1}$ in the upper troposphere, corresponding to about 20-50% of the total IWC there (Fig. 5 p,q,t). However, the IWC decrease in VEL2 is followed by an increase of IWC of a similar magnitude in the mixed-phase cloud regime at temperatures warmer

than -35°C, where the IC sedimentation velocity is restored to the reference value.

The IWP and ICNC pattern in VEL2 are a result of an unrealistic redistribution of ice mass and number concentration to lower levels (Fig. 5 q,v), while in SEED1r50 (Fig. 5z) the large, newly formed ICs quickly grow by vapour deposition to sizes large enough to precipitate and be removed from the atmosphere. Moreover, the convectively detrained ICs dominate ICNC at locations below about 250 hPa in our model (Gasparini and Lohmann, 2016). Including new freezing INP in the environment by either convection or heterogeneous IC nucleation only redistributes the same amount of vapour between more particles as explained in Gasparini and Lohmann (2016), leading to a decrease in IC radii. The ice water path (IWP) decreases by about 7% in SEEDr50 but only by about 1.5% in the VEL2 simulation (Fig. 4e). In SEED1r50 the effect of quickly sedimenting large ICs from cirrus levels to some extent also affects the ICs in mixed-phase clouds, leading to a small IWC decrease also at temperatures warmer than -35°C (Fig. 5k).

On the other hand, in VEL2 the redistribution of IWC and ICNC from the cirrus levels to warmer temperatures leads additionally to a mixed-phase cloud glaciation effect: the cloud droplet number concentration (CDNC) at temperatures between -35°C and 0°C therefore decreases at the expense of the additional increase in ICNC. This leads to a globally averaged liquid water path decrease by about 3%, CDNC drop by 1%, and ICNC increase by 7% (Fig. 4e and Fig. 6b). In VEL2 the ICs at cirrus levels fall faster and have therefore less time available for depositional growth, leading to a decrease in their size (Fig. 5aa), confirming the results of Muri et al. (2014)

The decrease in cirrus cloud cover changes the atmospheric diabatic heating, leading to an upper tropospheric cold anomaly of about 1°C in both simulations (Fig. 5 k,l,o). Interestingly, the location of the maximum anomaly and its vertical extent differ significantly between SEED1r50 and VEL2. The peak temperature decrease in SEED1r50 is concentrated in the tropical tropopause region, while VEL2 shows an elongated negative temperature anomaly extending to about 7 km altitude. The reason for this 5 km difference in the altitude of peak cooling is most likely related to the inability of seeding to influence the convectively formed and other liquid origin cirrus clouds, which dominate at temperatures warmer than -50°C (Voigt et al., 2016). On the contrary in the simulation VEL2 all cirrus ICs are affected by the increased sedimention regardless of their origin. Considerably larger TOA SW and LW radiative flux anomalies in SEED1r50 compared to VEL2 occurs at a higher altitude as well as the maximum radiative forcing anomaly (Fig. 4c). Moreover, in SEED1r50 the destabilisation of the upper troposphere leads to increased vertical velocities and increased tropical tropopause and stratospheric specific humidities. This implies higher cooling rates dominated by the LW emissivity of water vapour (Clough and Iacono, 1995), and explains part of the tropical tropopause and the stratospheric temperature signal.

### 3.2.2 Other cloud responses to seeding

The anomalies of cCRE are almost a factor of two larger than the net TOA balance anomalies (Fig. 2) or net TOA CRE anomalies (Fig. 4) as evaluated from the fixed SST setup, where the TOA radiative fluxes do not reach a new equilibrium. The additional diagnostics of liquid and mixed-phase CRE (Table 5) point at additional cloud responses to cirrus geoengineering

that exert a positive radiative forcing and thus weaken the effect of cirrus geoengineering. We note that the additional CRE decomposition is performed in fixed SST simulation setup which, however, leads to very similar cloud responses as in the corresponding MLO simulations.

5     The VEL2 simulation leads to a redistribution of ice from the cirrus to the lower lying mixed-phase regime, exerting a positive mixed-phase cloud forcing of about 0.5 W m$^{-2}$ (Fig. 5q and Table 5). Changes in vertical stability lead to an increase in midlevel convection and additionally contribute to the redistribution of ice. These changes are responsible for part of the increases in ICNC burden associated with the convectively detrained ICs (Fig. 4e and Fig. 5 v). The positive anomaly in RH at 5-10 km (Fig. 5g) is concentrated in and just above areas of vertical velocity increase (not shown), driven by changes in 10  vertical temperature gradients (Fig. 5o).

    Furthermore, in SEED1r50 an increase or intensification of convective activity, expressed by an 1.2% increase in globally averaged convective precipitation, leads to a drying of the tropical planetary boundary layer and lower troposphere and a decrease in liquid cloud cover (Fig. 5a). The cloud cover is directly related to RH (Sundqvist et al., 1989); its decrease therefore 15  leads to a cloud cover decrease, which decreases also the all-sky water content and CDNC (Fig. 6 a,c and Fig. 4e). This exerts a positive liquid CRE anomaly (Table 5), similarly to what was found in studies by Rieck et al. (2012) and Sherwood et al. (2014). Interestingly, the shift from homogeneous to heterogeneous ice nucleation leads to higher RH in the upper troposphere (Fig. 5f). The heterogeneous nucleation and growth timescale is several times longer than the homogeneous one (Köhler and Seifert, 2015), leading to slower vapour consumption by depositional growth on ICs at our optimal seeding concentration of 1 INP L$^{-1}$.

    The sedimentation of ICs into mixed-phase clouds leads to the IC growth by riming of supercooled cloud droplets, also known as the seeder-feeder mechanism (Politovich and Bernstein, 1995). The seeder-feeder mechanism, reinforced by the additional growth of ice crystals at the expense of supercooled cloud droplets (Wegener-Bergeron-Findeisen process (Storelvmo and Tan, 2015)) leads to a depletion of CDNC and LWC. A decrease of ICs sedimenting flux from cirrus in SEED1r50 com- 25  pared to REF therefore leads to an increase in CDNC and LWC in mixed-phase cloud regime. In the tropics, this process is contrasted by the large RH decrease (Fig. 5f), leading to a decrease in cloud cover (Fig. 5a), and consequently also a small and not significant decrease in all-sky CDNC and LWC in Fig. 6a.

    By seeding cirrus clouds only at night (simulation SEED1r50N) we target their warming LW CRE and obtain a similar 30  net TOA flux anomaly (-0.88 $\pm$ 0.36 W m$^{-2}$) without significantly perturbing the SW balance as in other simulations (Table 5). Despite obtaining a smaller annually averaged cCRE, the net radiative decrease at the TOA is similar to the one in the simulation SEED1r50 (Fig. 2c). SEED1r50N triggers only a small increase in convective activity (0.8% increase in convective precipitation compared with 2.8% in SEED1r50) and thus limits the drying of the boundary layer and the decrease of the liquid CRE (Table 5).

### 3.2.3  Cirrus cloud radiative effects and temperature

We separately diagnose cirrus cloud radiative effects from the two MLO simulations (SEED1r50, VEL2) to evaluate the regions of highest seeding effectiveness. Both scenarios produce similar net cCRE anomalies, which follow the climatological pattern of cirrus cloudiness and their radiative impacts at the TOA, having the largest effect in the warm pool region, in storm tracks, and over orographic barriers (Fig. 7 a,b). In SEED1r50 the anomaly pattern shows an even more pronounced impact over mountain regions and the tropical warm pool than in VEL2, corresponding to regions dominated by homogeneously nucleated ICs (Gasparini and Lohmann, 2016).

Temperature anomalies in general follow the anomalies in cCRE and are about 30-40% larger over land than over the ocean (Fig. 7 d,e). Yet, the temperature anomaly in the Pacific warm pool area is an exception to this general trend, which needs to be addressed in future studies. Moreover, both scenarios have larger responses in high latitudes, with a cooling of almost 2°C in the annual average, similar to findings of Storelvmo et al. (2014) and Muri et al. (2014). The globally average surface temperature decrease is about 30% larger in VEL2 (-0.68 $\pm$ 0.13°C) compared with SEED1r50 (-0.49 $\pm$ 0.09 °C). The difference is likely explained by the larger surface forcing in VEL2 compared to SEED1r50 (-0.86 and -0.74 W m$^{-2}$, respectively), which is the result of a lower altitude of the maximum diabatic cooling anomaly in VEL2 (Fig. 5o). The radiative response is further amplified by changes in precipitable water, acting primarily in the tropics (Fig. 4e), and the sea ice feedback in high latitudes (not shown).

Both cCRE and temperature anomalies have a strong seasonal cycle (Fig. 7 c,f). The cooling is particularly pronounced in the winter hemisphere, and can exceed 3°C in the Arctic or 2°C in the Antarctic winter. The high-latitude cooling is a combination of atmospheric geoengineering and the ice-albedo feedback. The two scenarios exhibit remarkably similar zonally averaged temperature responses with no significant differences.

Interestingly, the SEED1r50N scenario leads to a 0.1°C larger globally averaged surface temperature cooling effect with a similar seasonality (Fig. 8a). The SEED1r50N scenario is more effective in the tropical deep convective areas, which probably originates from differences in the response of liquid clouds (in SEED1r50 the liquid clouds have a positive CRE anomaly, Table 5). The mid and high latitude temperature anomaly pattern likely reflect changes in extratropical interannual climate variability modes. Most notably, we observe a significant Arctic cooling and warming in the northern hemispheric midlatitudes (Fig. 8 b,c), associated with a pressure decrease over the Arctic and increase over most of the midlatitudes (not shown), resembling a positive Northern Annular Mode temperature signal (Thompson and Wallace, 2000).

### 3.3  Alternative modelling strategies to increased sedimentation velocity

The INP seeding setup, as opposed to the increased sedimentation velocity setup, does not allow modifications of lower lying liquid origin cirrus clouds, which are mainly dynamically controlled anvils of convective clouds (Penner et al., 2015). Such clouds most likely cannot be influenced by seeding as they are less sensitive to changes in microphysics. The temperature of

the boundary between liquid origin and in situ cirrus is also latitudinally dependent: a study by Jensen et al. (2015) suggested this boundary to be rather close to -70°C in the tropics, with -50°C being more representative for the midlatitudes (Voigt et al., 2016; Wernli et al., 2016).

5    In order to bridge the gap between increased sedimentation velocity and seeding simulations we performed an additional simulation using a lower temperature threshold of -50°C to modify prevalently in situ formed cirrus (VELmax-50). However, a large proportion of cirrus clouds that strongly influence the global radiative budget resides in the temperature range between -35°C and -50°C. The CRE of cirrus clouds colder than -50°C is only 1.7 W m$^{-2}$ as compared to the 4.4 W m$^{-2}$ for all cirrus clouds according to our model (Table 3). Therefore, we need to set the sedimentation velocity of ICs at temperatures lower than 
10   -50°C to the maximum allowed by the model (2 m s$^{-1}$) to obtain a similar, but still with -0.4°C significantly smaller globally averaged cooling effect. Simulation VELmax-50 approximately reproduces the SEED1r50 cloud cover anomaly pattern (Fig. 5 a,d,e) and upper tropospheric temperature anomalies (Fig. 5 k,n,o). However, in simulation VELmax-50 the IWC at temperatures warmer than -50°C increases substantially (Fig. 5s), leading to an increase in the ICNC and IWP (Fig. 4e). The ICNC increases even at the tropical tropopause (Fig. 5 x), which is a microphyiscal response to a decrease of temperature for up to 
2°C in the same region (Fig. 5 n) and can be eliminated by nudging the temperature in the seeded simulation to the reference simulation values (not shown).

Interestingly, VELmax-50 exerts a smaller radiative flux and temperature perturbation but a fast precipitation response comparable to the one in the VEL2 simulation (Fig. 4 a,b). Both the large fast precipitation response and the smaller temperature decrease lead to an overall net small and not statistically significant precipitation increase in the MLO simulation setup, differ-
ently from other simulations (Fig. 4d).

We additionally performed an arguably more physical simulation, in which the IC sedimentation velocity is increased for both ICs at temperatures warmer and colder than -35°C (VEL2all, see Table 1). VEL2all leads to strikingly similar radiative and precipitation responses as in the VEL2 and SEED1r50 simulations, inducing a slightly larger surface cooling effect (Fig.
4d). Interestingly, in VEL2all IWC decreases throughout the atmosphere only in the extratropics. Its tropical mid-tropospheric IWC increase is likely caused by a convective activity increase (convective precipitation increases by 1.4%), resulting in a similar RH anomaly peak at 7-10 km altitude as in the VEL2 simulation (Fig. 5 g,h). These changes in deep and mid-level convection, which lead to a large number of small ICs are also responsible for an 8% increase in ICNC burden despite a 6% decrease in IWP. The surprising result can be explained by the parametrization of the size of the detrained ICs, which assumes
an IC radius of about 10-20 $\mu$m (Boudala et al., 2002), distributing the detrained IWC over a large number of ICs and is part of the reason for the ICNC increase pattern in mixed-phase cloud conditions (Fig. 5w). Moreover, the freezing in mixed-phase clouds in our model occurs only rarely in situ onto dust or black carbon aerosols and is largely affected by the sedimented ICs from cirrus levels, initiating a seeder-feeder type of IC growth.

Yet, neither VEL2all nor any other cirrus geoengineering method with increased IC sedimentation velocity can reproduce
the magnitude of cloud cover and temperature changes induced by seeding with effective INPs. Our simulations show that

idealised cirrus seeding simulations by means of increased sedimentation velocity are not a good proxy for cloud macro- and microphysical changes. Nevertheless, simulations with increased IC sedimentation velocities can still provide useful information for some climatic responses (e.g. surface temperature, precipitation) to cirrus seeding.

## 4 Conclusions

We studied the climatic responses to cirrus seeding and to increased sedimentation velocity of ice crystals in cirrus clouds. In general, the increased sedimentation velocity simulation (VEL2) leads to qualitatively similar responses compared to cirrus seeding with large INPs (SEED1r50): a decrease in cloud cover and ice water content at cirrus levels, and a temperature decrease throughout the troposphere. However, in VEL2 the IC sedimentation velocity is abruptly set back to the standard one computed by the model at temperatures warmer than -35°C, leading to a redistribution of ICs and IWC from cirrus to underlying mixed-phase clouds, which is not observed in SEED1r50. Our general findings therefore indicate that increasing sedimentation velocity is a good proxy for cirrus seeding surface climate responses, while it cannot reproduce the complex cloud macro- and microphysical responses. The additional simulations with increased sedimentation velocity for all ice crystals (VEL2all) or only for those at T<-50°C (VELmax-50) also cannot reproduce all the seeding signals from the seeding scenario. An accurate evaluation of atmospheric changes of cirrus thinning therefore requires the implementation of a cirrus microphysics scheme that is able to simulate the competition between homogeneous and heterogeneous ice crystal nucleation.

The maximum impact on TOA radiative fluxes by the increase of ice crystal sedimentation velocity is -2.2 $\mathrm{W\,m^{-2}}$, which corresponds to about half of the cirrus CRE and half of the radiative forcing of doubling of $CO_2$. The maximum impact of seeding with effective INPs is, on the other hand, only about -1 $\mathrm{W\,m^{-2}}$ or 20-25% of cirrus CRE, which is achieved by injecting large ice nucleating particles of 50 $\mu$m radius in the SEED1r50 simulation.

A large part of the cirrus geoengineering induced negative CRE is counteracted by decreases in liquid clouds in the SEED1r50 simulation in response to increased convective activity. In addition, the redistribution of ice to lower levels in VEL2 leads to a positive mixed-phase cloud CRE anomaly. As shown in Fig. 2, implementing seeding only during night leads to a comparable cirrus CRE and net radiative anomaly signal, without any significant counteracting effect from liquid or mixed-phase clouds (Table 5). Interestingly, such a seeding strategy leads to no significant fast precipitation response and smaller changes in IWP, ICNC, LWP, and CDNC compared to SEED1r50.

The mean climatic responses of both the SEED1r50 and VEL2 simulations in terms of radiative fluxes are roughly similar. Yet, the microphysical differences between the two setups lead to a different vertical cooling patterns and a 30% larger surface temperature response in the VEL2 simulation. As the surface temperature response pattern in the annual and seasonal averages is similar, we expect to achieve the same amount of surface cooling by a smaller increase of IC sedimentation velocity.

Moreover, precipitation responds to both geoengineering strategies in a similar way. The fast responses to seeding yield a ~1% increase in precipitation, while the slow, temperature driven response in the mixed layer ocean simulations leads to a 0.5% decrease.

The SEED1r50N strategy shows, despite seeding only in the night, a slightly larger surface temperature response and a twice as large precipitation decrease which follows more closely the temperature dependence of the Clausius-Clapeyron relation (7% precipitation decrease per 1°C cooling). SEED1r50N therefore seems to be, not considering its unlikely technical implementation, our most appealing seeding simulation due to minimal climatic and microphysical responses outside the cirrus regime. We note that a seeding strategy limited to areas of highest seeding effectiveness (where the cCRE anomalies after seeding are the largest, as shown by Fig. 7a), might significantly decrease the mass of seeded material while exerting a roughly similar climatic forcing.

The seeding effectiveness does not only depend on the seeding INP properties, but also on the relative frequency between both the in situ and liquid origin cirrus and homogeneously vs. heterogeneously in situ formed cirrus, which may differ between the model and observations and between different models. We also expect the cirrus seeding effectiveness to be dependent on the amount of background aerosol available for both homogeneous or heterogeneous freezing (Zhou and Penner, 2014). Furthermore, the effectiveness of cirrus seeding measured in terms of radiative anomalies is highly dependent on the cirrus CRE and consequently also on model parameters that have a large effect on cirrus optical properties.

Ice cloud radiative effects are poorly constrained by observations on the global scale and rarely explicitly diagnosed from modelling studies. We suggest to invest more resources in understanding the cirrus cloud formation mechanisms and radiative effects at high temporal resolutions in order to better constrain CRE effects. Until then, we propose to not only state the radiative impact in terms of $W\,m^{-2}$ achieved by cirrus geoengineering simulations (either by injection of seeding INPs or by increasing ice crystal sedimentation velocities) but also the fraction of the total cirrus cloud radiative effect that is eliminated by cirrus geoengineering.

## 5 Code availability

## 6 Data availability

The data from the model simulations are available from the authors upon request.

*Author contributions.* TEXT

*Competing interests.* TEXT

*Disclaimer.*

*Acknowledgements.* The simulations were performed on Daint cluster of the Swiss National Supercomputing Center (project s431) and on the Euler ETHZ computational cluster. We would like to thank David Neubauer for his valuable comments and help on technical issues. We thank Marina Dütsch, Katty Huang, and Robert David for suggesting improvements to the manuscript. The data from the model simulations are available from the authors upon request.

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

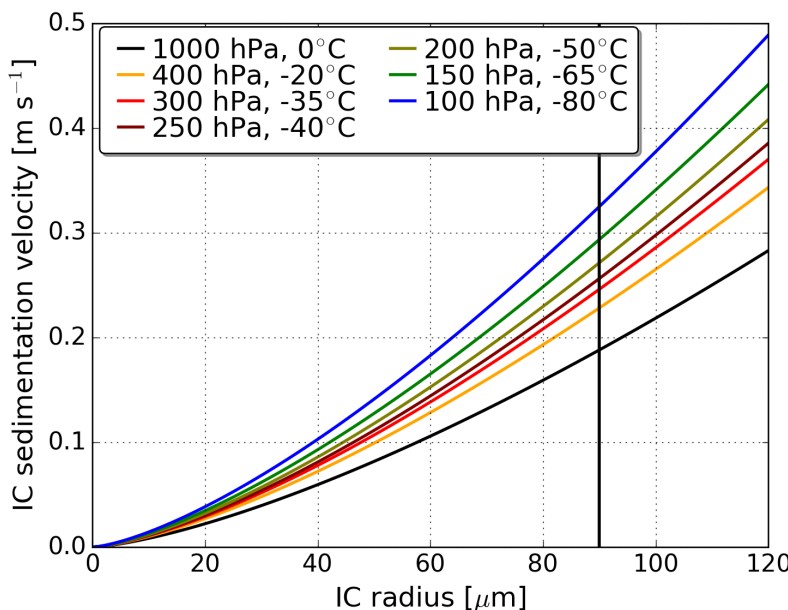

**Figure 1.** Ice crystal sedimentation velocities as a function of IC radius for the selected atmospherically relevant conditions in ECHAM-HAM (Spichtinger and Gierens, 2009). The black vertical line represents the maximum radius ice crystals can have before they are transferred to the precipitating snow category.

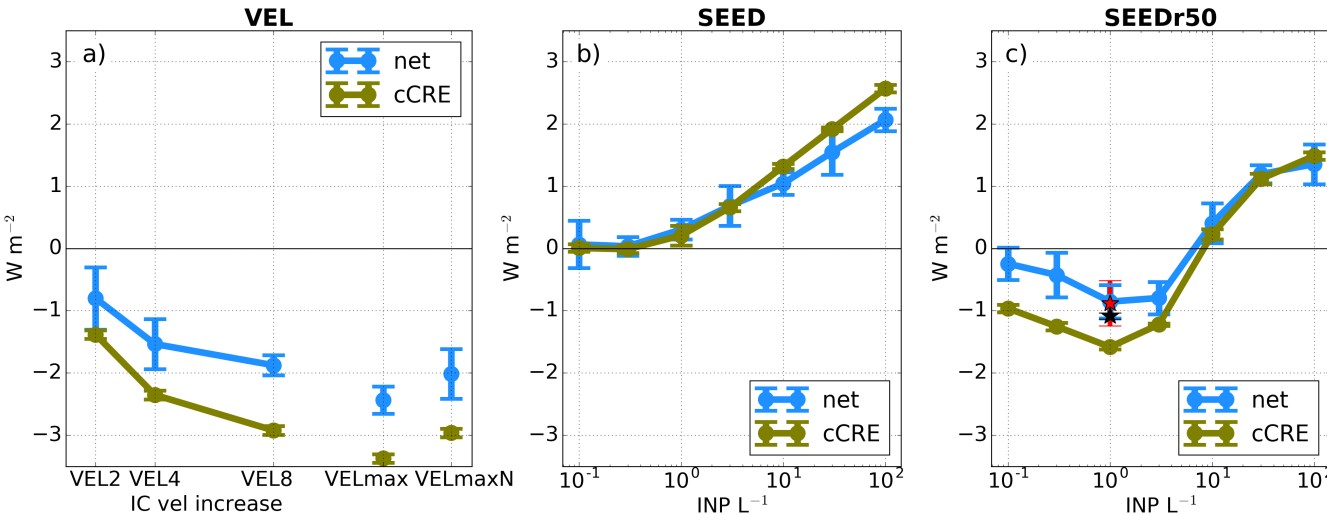

**Figure 2.** 5-year TOA anomalies of net radiative fluxes (NET) and net cirrus cloud radiative effects (cirrus CRE) from fixed SST simulations of seeding with increased sedimentation velocities (a). b) shows net radiative fluxes and cirrus CRE for seeding simulations with different 0.5 $\mu$m sized INPs, while c) shows the equivalent for seeding with 50 $\mu$m INPs. The stars in c) show results from the SEED1r50N simulation with seeding performed only at night where the red star represents the net anomaly and the black one cirrus CRE anomaly. The error bars represent $\pm$ 2 standard deviations.

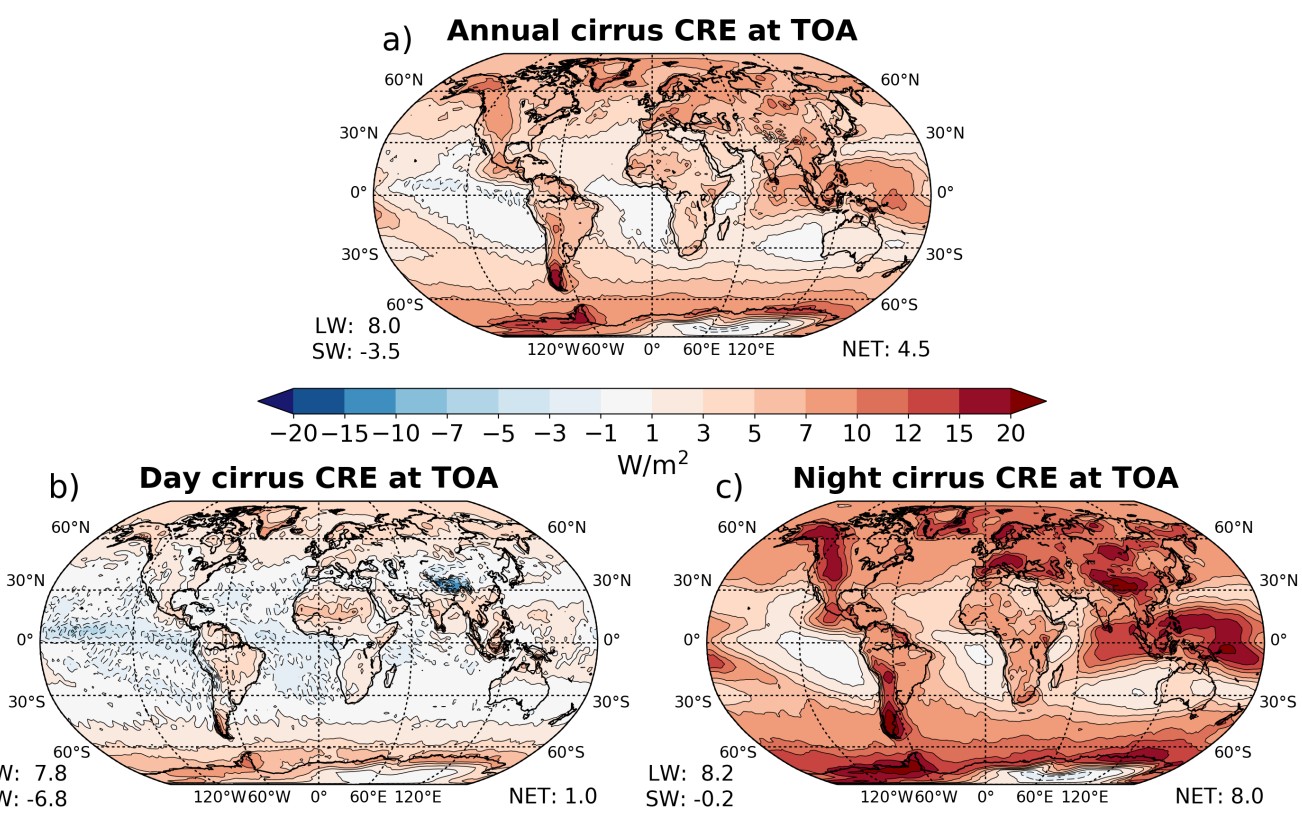

**Figure 3.** 5-year all-sky TOA anomalies of net cirrus cloud radiative effects (cirrus CRE) for the reference (unseeded) fixed SST simulation in annual average (a) and when computed only during day (b) and night (c). The day/night definition is based on the solar zenith angle, where day includes all gridboxes with the sun above the horizon. Panel (a) is reproduced with slight modifications from Gasparini and Lohmann (2016).

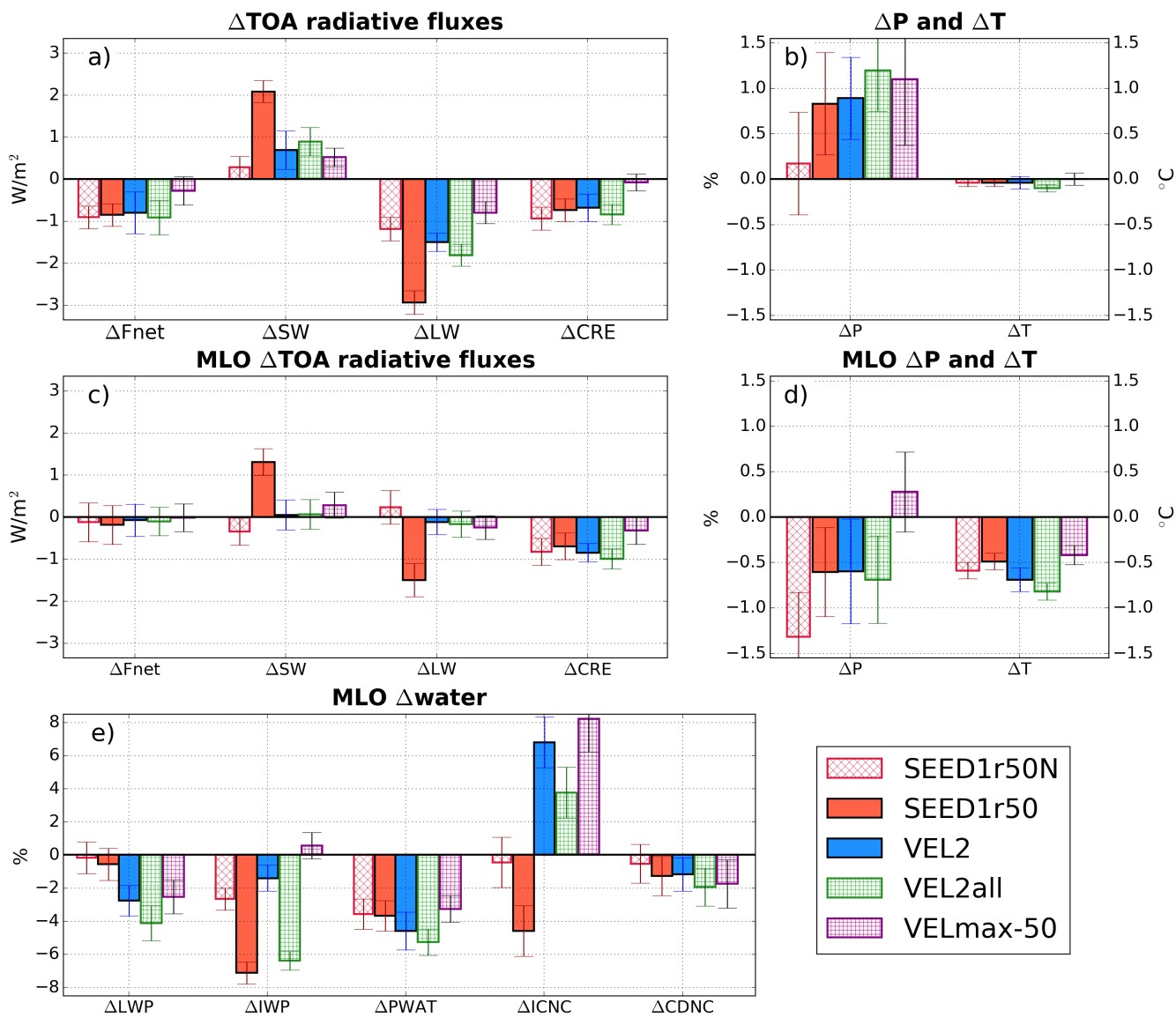

**Figure 4.** Annually averaged anomalies for top-of-the-atmosphere (TOA) energy fluxes (a,c), precipitation (P), and temperature (T) (b,d) and selected quantities of the hydrological cycle (e): liquid water path (LWP), ice water path (IWP), precipitable water (PWAT), ice crystal number concentration (ICNC), cloud droplet number concentration (CDNC). a) and b) show anomalies from fixed SST simulations, c), d), and e) from mixed layer ocean (MLO) simulations. The error bars represent the $\pm\,2$ standard deviation range.

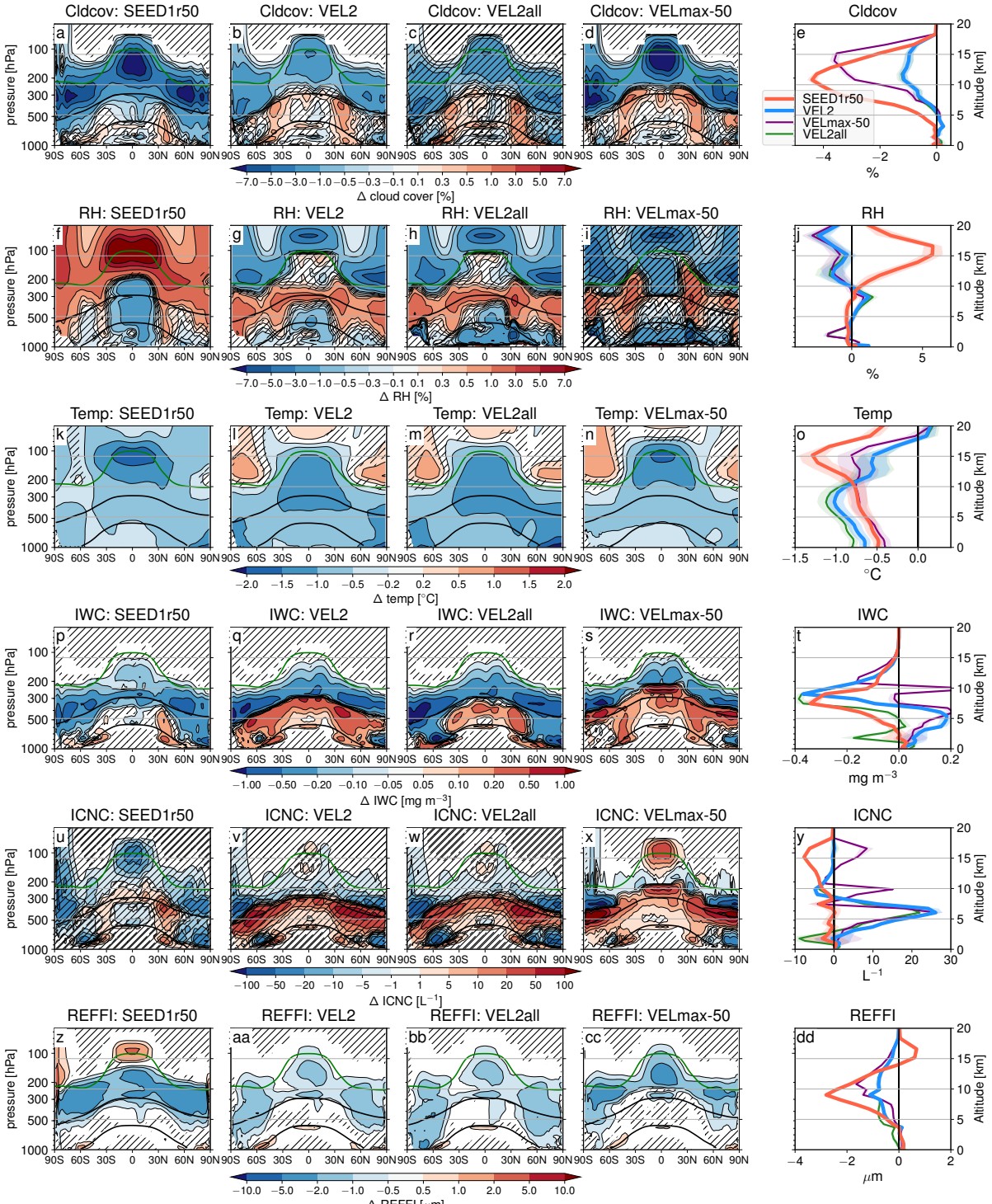

**Figure 5.** Annually averaged anomalies of cloud cover (Cldcov, a-e), relative humidity with respect to liquid water (RH, f-j), temperature (Temp, k-o), all-sky ice water content (IWC, p-t), all-sky ice crystal number concentration (ICNC, u-y), and all-sky ice crystal effective radius (REFFI, z-dd) for the SEED1r50 and VEL2 MLO simulations (see Table 1). The green curve represents the tropopause, and the black curves the -35°C and the 0°C isolines. The hatching is applied for anomalies not significant at the 95% confidence level. On the right hand the anomalies are averaged over latitude and longitude.

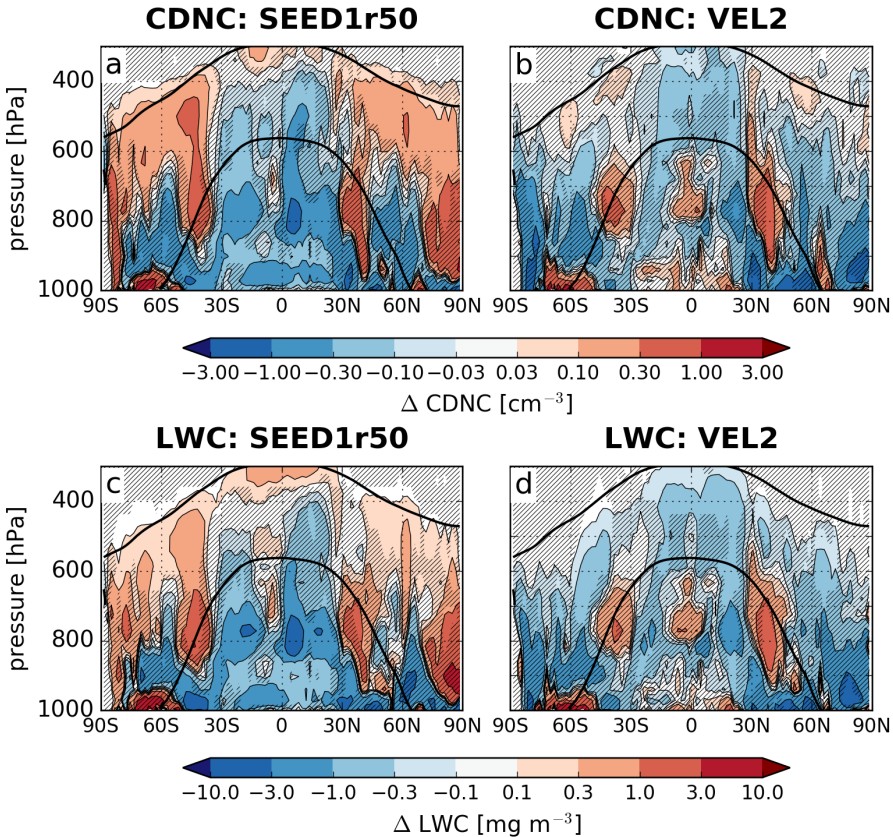

**Figure 6.** Annually averaged all-sky anomalies of the cloud droplet number concentration (CDNC, a,b) and liquid water content (LWC, c,d) for the SEED1r50 and VEL2 fixed SST simulations (see Table 1). The black curves are the -35°C and the 0°C isolines. The hatching is applied for anomalies not significant at the 95% confidence level.

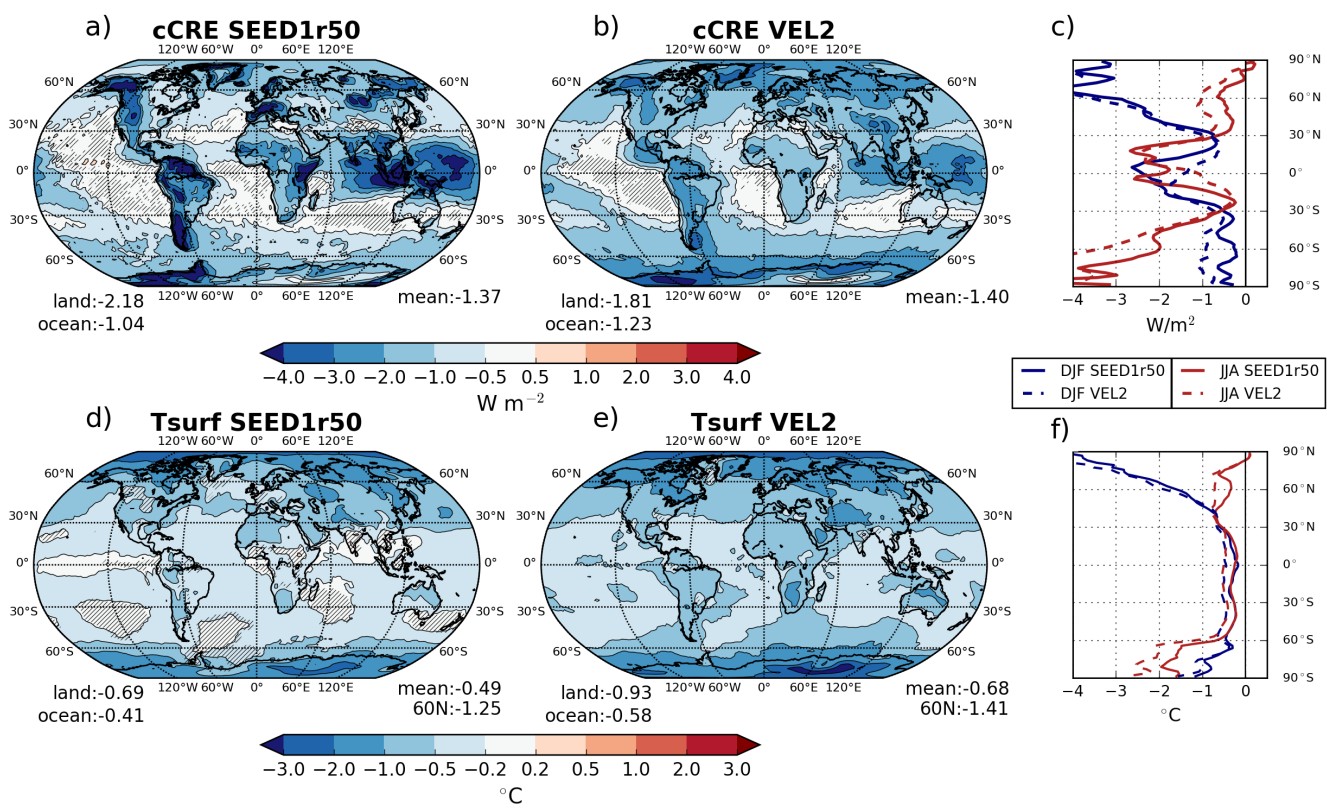

**Figure 7.** Annually averaged anomalies of cirrus cloud radiative effect (cCRE, a,b) and surface temperature (Tsurf, c,d) from the SEED1r50 and VEL2 MLO simulations. The hatching is applied for anomalies not significant at the 95% confidence level. Panels c) and f) show the respective annual zonal averages for DJF (blue) and JJA (red).

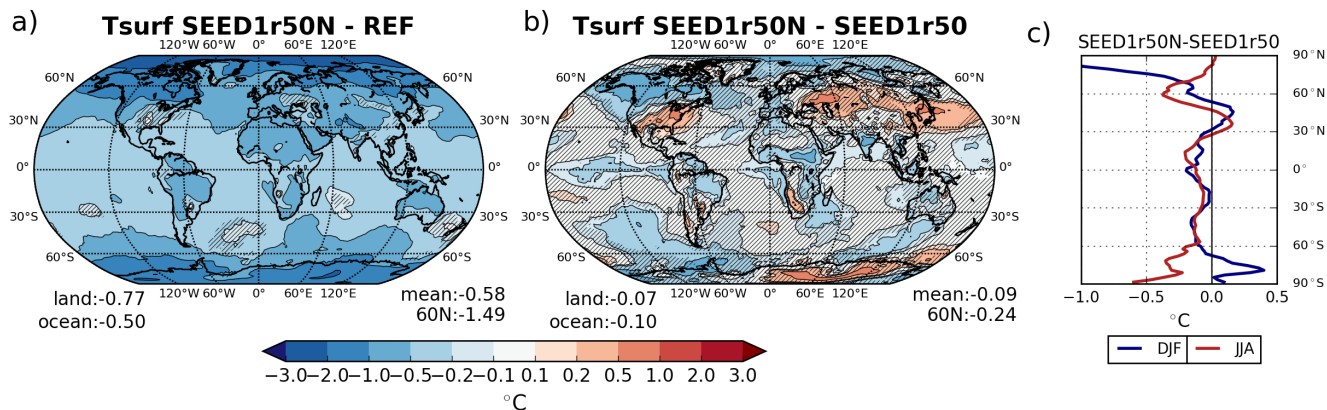

**Figure 8.** Annually averaged anomalies of surface temperature (Tsurf) for the SEED1r50N simulation with respect to REF (a) or SEED1r50 (b). The hatching is applied for anomalies not significant at the 95% confidence level. Panel c) shows the respective annual zonal average anomalies of SEED1r50N with respect to SEED1r50 for DJF (blue) and JJA (red). All simulations are performed in the MLO setup.

**Table 1.** Simulation terminology and their respective cirrus geoengineering method.

| Simulation | Sim. length [y] | IC sedimentation | seed INP conc. [$L^{-1}$] | seed INP radius [$\mu$m] |
|---|---|---|---|---|
| **fixed SST** | | | | |
| REF | 10 | / | / | / |
| VEL2 | 10 | 2 x ref | / | / |
| VEL4 | 5 | 4 x ref | / | / |
| VEL8 | 5 | 8 x ref | / | / |
| VELmax | 5 | set to 2 m/s | / | / |
| VELmaxN | 5 | set to 2 m/s at night | / | / |
| VEL2all | 5 | 2 x ref (all ICs) | / | / |
| VELmax-50 | 5 | 2 x ref (at T<-50°C) | / | / |
| SEED0.1 | 5 | / | 0.1 | 0.5 |
| SEED0.3 | 5 | / | 0.3 | 0.5 |
| SEED1 | 5 | / | 1 | 0.5 |
| SEED3 | 5 | / | 3 | 0.5 |
| SEED10 | 5 | / | 10 | 0.5 |
| SEED30 | 5 | / | 30 | 0.5 |
| SEED100 | 5 | / | 100 | 0.5 |
| SEED0.1r50 | 5 | / | 0.1 | 50 |
| SEED0.3r50 | 5 | / | 0.3 | 50 |
| SEED1r50 | 10 | / | 1 | 50 |
| SEED3r50 | 5 | / | 3 | 50 |
| SEED10r50 | 5 | / | 10 | 50 |
| SEED30r50 | 5 | / | 30 | 50 |
| SEED100r50 | 5 | / | 100 | 50 |
| SEED1r50N | 10 | / | 1 at night | 50 |
| SEED1r5 | 5 | / | 1 | 5 |
| SEED1r10 | 5 | / | 1 | 10 |
| SEED1r20 | 5 | / | 1 | 20 |
| **Mixed layer ocean (MLO)** | | | | |
| REF | 50 | / | / | / |
| SEED1r50 | 50 | / | 1 | 50 |
| SEED1r50N | 50 | / | 1 at night | 50 |
| VEL2 | 50 | 2 x ref | / | / |
| VEL2all | 50 | 2 x ref (all ICs) | / | / |
| VELmax-50 | 50 | 2 x ref (at T<-50°C) | / | / |

**Table 2.** Cirrus geoengineering CRE anomalies in W m$^{-2}$ with respect to REF (first column) or with respect to the simulation with two times smaller ICs sedimentation velocities (second column) for fixed SST simulations. The last column represents the relative fraction of the cirrus cloud radiative effect (cCRE) anomaly with respect to the remaining cCRE.

|        | $\Delta$cCRE [W m$^{-2}$] | $\Delta$ remaining cCRE [W m$^{-2}$] | $\Delta$ remaining cCRE [%] |
|--------|---------------------------|--------------------------------------|-----------------------------|
| VEL2   | -1.43                     | -1.43                                | -33%                        |
| VEL4   | -2.40                     | -0.97                                | -33%                        |
| VEL8   | -2.98                     | -0.58                                | -30%                        |

**Table 3.** Net cirrus cloud radiative effects (cCRE) from the fixed SST REF simulation of all clouds at temperatures colder than the one stated in the left column. The right column represents the percentage contribution to the total cCRE.

| Temp [$^{\circ}$C] | cCRE [W m$^{-2}$] | percentage [%] |
|--------------------|-------------------|----------------|
| -35                | 4.35              | 100            |
| -40                | 3.42              | 79             |
| -45                | 2.49              | 57             |
| -50                | 1.73              | 40             |
| -55                | 1.22              | 28             |
| -60                | 0.83              | 19             |
| -65                | 0.53              | 12             |
| -70                | 0.34              | 8              |

**Table 4.** Top-of-the-atmosphere net radiative balance ($F_{net}$) anomalies in W m$^{-2}$ for cirrus seeding with 1 INP L$^{-1}$ with varying INP radius and the $\pm$ 2 standard deviation range for fixed SST simulations.

|                              | SEED1           | SEED1r5         | SEED1r10         | SEED1r20         | SEED1r50         |
|------------------------------|-----------------|-----------------|------------------|------------------|------------------|
| $\Delta F_{net}$ [W m$^{-2}$] | 0.30 $\pm$ 0.30 | 0.01 $\pm$ 0.44 | -0.03 $\pm$ 0.41 | -0.46 $\pm$ 0.14 | -0.85 $\pm$ 0.40 |

**Table 5.** Top-of-the-atmosphere net cloud radiative effect anomalies of cirrus geoengineering with the individual contributions from cirrus clouds (cCRE, for temperatures <-35$^{\circ}$C), mixed-phase clouds (mpCRE, -35$^{\circ}$C < T < 0$^{\circ}$C), and liquid clouds (liqCRE, T > 0$^{\circ}$C) for the VEL2, SEED1r50, and SEED1r50N fixed SST simulations. The radiative anomalies are further divided into their LW and SW components (shown in parenthesis).

| simulation | $\Delta$liqCRE (LW, SW) [W m$^{-2}$] | $\Delta$mpCRE (LW, SW) [W m$^{-2}$] | $\Delta$cCRE (LW, SW) [W m$^{-2}$] | $\Delta$totCRE (LW, SW) [W m$^{-2}$] |
|------------|--------------------------------------|-------------------------------------|------------------------------------|--------------------------------------|
| VEL2       | 0.09 $\pm$ 0.38 (0.02, 0.07)         | 0.41 $\pm$ 0.12 (0.82, -0.42)       | -1.43 $\pm$ 0.06 (-2.02, 0.60)     | -0.84 $\pm$ 0.41(-1.47, 0.63)        |
| SEED1r50   | 0.96 $\pm$ 0.25 (0.00, 0.96)         | 0.15 $\pm$ 0.10 (0.24, -0.10)       | -1.63 $\pm$ 0.03 (-2.47, 0.84)     | -0.82 $\pm$ 0.31 (-2.90, 2.08)       |
| SEED1r50N  | 0.15 $\pm$ 0.14 (0.01, 0.14)         | 0.18 $\pm$ 0.18 (0.13, 0.04)        | -1.06 $\pm$ 0.03 (-1.07, 0.02)     | -0.95 $\pm$ 0.20 (-1.21, 0.26)       |