# Peer review of "Is increasing ice crystal sedimentation velocity in geoengineering simulations a good proxy for cirrus cloud seeding?"

_Atmospheric Chemistry and Physics, 2016_

## Referee Comment (RC1) · Anonymous Referee #1 · 10 Jan 2017

**Review of "Is increasing ice crystal sedimentation velocity in geoengineering simulations a good proxy for cirrus cloud seeding? By Gasparini et al.**

This is a well written paper. I have a few suggestions to help to clarify it, however.

Page 2, Sentence lines 19, 20: The idea that cirrus seeding does not influence the IC that nucleated at temperatures between 0 and -35 degrees seem wrong, since you state in abstract that ice seeding modifies liquid clouds.

Page 3, Line 27: what is the radius associated with a 2 m/s sedimentation velocity. This must be outside of the normal realistic range as shown in Fig 1.

Page 4, Line 6, 7: please explain why you consider this a "problem". Were there inaccuracies within the calculation?

Page 4, line 28: replace: "decreases for about" to "decreases by about"

Page 5,line 6: "15-20% smaller radiation effect decrease:  smaller than VELmax, but not smaller than VEL8. Please be specific.

Page 5, line 5:  this seems to say that the total CRE is 1 W/m2 during the day. Really?

Page 5, line 6-8: I'm not sre what you are saying here. cCRE in Table 3 is 4.35 W/m2. Is this 50% smaller than that in REF? Please add values for REF in Table 3.

Page 6 line 22: a lifetime of a couple of hours seems small. Are there observations to examine whether this number is reasonable?

Page 7 line 1, 2:  In the model description section you need to specify how cloud cover is calculated. Is it just related to relative humidity?

Page 7:  lines 4 – 6: "where the IC sedimentation velocity is restored to the reference values." Doesn't this assumption essentially change the removal treatment for cirrus? Is this a common assumption in models that alter the sedimentation velocity to simulate geoengineering?

Page 7, line 23: "in Vel2 all cirrus IC sediment regardless of their origin". So you are not applying the 2 m/s sedimentation only to the added INP for geoengineering?

Page 8 lines 3,4:  Are you surmising that these changes are responsible for increases in ICNC due to detrainment? i.e. you do not show this, so how do you know?

Page 8, line 8, 9: Why does an increase in convective activity lead to a decrease in CDNC?

Page 8, line 13-15: Why is CDNC higher in mixed phase cloud?

Page 8, line 31: "T anomalies follow cCRE anomalies" Not in general true, since you do not see warm pool anomalies.

Page 10: line 10: Why do change in tropical convection lead to a large number of small IC and an increase in ICNC?

Page 10, line 14: replace "on some climatic" with "for some climatic"

Page 11, line 24 – 26: Zhou and Penner, JGR, 2014 show different model assumptions that are used to describe the number of homogeneous and heterogeneous particles.

Page 11, line 28: what is the "inhomogeneity parameter for ice clouds"

Page 11: last line: what does "to which fraction of the total cirrus CRE this radiative anomaly corresponds" mean? The fraction due to liquid or mixed phase clouds?

Figure 2: Please explain why CRE is not the same as the net (all sky) results. Also, why does it switch from higher than net to lower than net?

Table 1: please write out what MLO means within the table (as well as in text).

Table 5: Please add a description of how ΔmpCRE, ΔcCRE, ΔliqCRE are computed within text.

---

## Referee Comment (RC2) · Anonymous Referee #2 · 12 Jan 2017

This is a nice and straightforward, if not exactly Earth shattering, modeling study of geoengineering in the form of cirrus seeding. The study includes highly relevant testing of the usefulness of the approach of mimicking cirrus seeding by increasing ice crystal fall speed. The latter has been proposed as an experiment in a new set of GeoMIP simulations, so it is important to understand in what respects this is a useful proxy for explicitly simulating the shift from homogeneous to heterogeneous nucleation that (at least in theory) occurs in response to seeding with INP.
I have a few major/substantial comments, and numerous minor comments which are listed below, and which are intended to help the authors improve the readability of the manuscript. Once the major/minor comments have been addresses, I believe the

manuscript will be suitable for publication in ACP.

Major comments:

- The authors conclude that the simulation that includes seeding at nighttime only is the most "appealing". To me, that simulation is a purely academic exercise, because in practice it would be impossible to only seed during night-time. It would take time to build up the right seeding INP concentration, and obviously one could not make the particles magically disappear at sunrise. I understand the desire to minimize the SW radiative effect, as well as the effects of increased convection, but I don't understand why the study doesn't include simulations with only high-latitude seeding (and fall speed increases). Seeding only high latitudes would, as opposed to the entire globe at night-time, potentially be possible in practice, and should achieve many of the same advantages. Considering the very large particles that appear to be most favorable seeding INP in this model, it would also be advantageous to seed much smaller areas to reduce the total mass required.

- The simulations with increased fall speed are, while in some respects useful, deeply unphysical. I missed a discussion of this in the manuscript. Specifically, I have problems with the sudden drop in fall speed that all ice crystals will experience as soon as they fall through the -35 degree isotherm. This would naturally lead to an accumulation of ice at mixed-phase levels, which is exactly what can be seen in Fig. 4. The authors seem to attribute this to mid-level convection. This is one important reason why increased fall speed is an imperfect proxy for actually simulating cirrus cloud microphysics.

Minor comments:

- Make sure you're consistent in your use of the abbreviations IC and INP - they are used to represent both the plural and singular forms of the nouns.

- Page 4, Line 20: "precipiation"–>"precipitation"

- Page 5, "for"–>"by"

- Page 5, Line 31-31: inappropriate referencing

- Page 7, Line 11: suggest replacing "affects also part of" with "to some extent also

affects"

- Page 7, Line 19: suggest replacing "temperature decrease anomaly" with "negative temperature anomaly"

- Page 11, Line 15: "twice larger"–> "twice as large"

- Figure 4: Why are ice crystal numbers and sizes not included in this figure? They seem like very important microphysical variables to include.

- Figure 5: Are these in-cloud or grid-box values?
* * *

---

## Referee Comment (RC3) · Anonymous Referee #3 · 17 Jan 2017

This is a manuscript on evaluating a simplified geo-engineering simulation strategy against the more comprehensive microphysical model treatment. The authors performed both fixed-SST and mixed-layer ocean simulations and investigated the changes in cloud properties for both ice and liquid phases, and the associated changes in cloud radiative forcing, surface temperature, and precipitation, due to using different geo-engineering model setups.

The manuscript is clearly written and well organized. The designed experiments and analysis are comprehensive. The conditionally sampled/calculated CRE changes are interesting and informative. I only have several questions that need clarification and some minor suggestions for better readability. Below please find my specific comments:

1. Page 2, Line 20: Cirrus seeding affects the sedimentation of ice crystals from cirrus clouds to mixed phase/warm clouds, the cloud glaciation, and therefore the ice supersaturation. Maybe change it to "cannot directly influence"?

2. Page 3, Line 12: What is the model time step? This is related to the question below about the numerical treatment of ice sedimentation.

3. Page 3, Line 26-27: If the model time step is about 20min, an ice crystal with a sedimentation velocity larger than 1m/s will fall through a 800m thick model layer (800m/1200s=0.67m/s). How does the model treat this (violation of CFL condition)?

4. Page 4, Line 5-6: What is the initial size of the nucleated ice particles? If not the size of the seeded INP, is it determined by the parameterization or by explicit microphysical calculation?

5. Page 4, Line 12: Does the seeded INP immediately freeze at RHi=105% when T<-35C? How do you consider the competition between the homogeneous freezing of solution droplets, heterogeneous freezing of natural dust, and heterogeneous freezing of the seeded INP?

6. Page 4, Line 13-14: Do you avoid the INP seeding in anvil clouds? Or in-situ ice nucleation doesn't happen in anvil clouds?

7. Page 5, Line 24-25: I don't quite understand this. Do you mean with the same amount of ice crystal mass?

8. Page 8, Line 27: The signal over topography seems pretty strong. Does the model consider the impact of orographic waves on ice nucleation?

9. Page 9, Line 28-29: This result is very interesting, but it's very likely model-dependent. Do you have extinction output as well? Would be nice to show the forcing efficiencies (&CF/&EXT) in table 3 as well.

10. Conclusion: Many points are made in the conclusion part and to me, they are a little bit scattered (very useful information though). I would recommend the authors to make it more compact and concise. Maybe in the order of 1) statement of general findings; 2) differences in microphysical responses; 3) differences in CREs; 4) differences in temperature and precipitation response; . . .?

---

## Author Comment (AC1) · 23 Mar 2017

**Response to reviewers**

We thank reviewers for their comments and suggestions. Please find the point-by-point reply (normal font) to the reviewer's comments (in bold) below. Blue text refers denote the changed text.

**REVIEWER 1**

**This is a well written paper. I have a few suggestions to help to clarify it, however.**

**Page 2, Sentence lines 19, 20: The idea that cirrus seeding does not influence the IC that nucleated at temperatures between 0 and -35 degrees seem wrong, since you state in abstract that ice seeding modifies liquid clouds.**

Thanks for pointing that out, changed in text, page 2 line 30.

**Page 3, Line 27: what is the radius associated with a 2 m/s sedimentation velocity. This must be outside of the normal realistic range as shown in Fig 1.**

According to the sedimentation velocity parametrization used in our model (Spichtinger et al., 2009), we need an ice crystal of about 1 μm size to reach such velocities. Please see the attached plot with the extended IC size range.

Added in the manuscript, page 4, line 18.
Only an IC of about 1 mm size would fall with a velocity of 2 m s$^{-1}$.

[Figure]

**Page 4, Line 6, 7: please explain why you consider this a "problem". Were there inaccuracies within the calculation?**

We call the "problem" the limiting factor to effective cirrus seeding (i.e. seeding which brings a cooling effect). Indeed, that is our internal jargon and has been removed from the text.

**Page 4, line 28: replace: "decreases for about" to "decreases by about"**

replaced

**Page 5,line 6: "15-20% smaller radiation effect decrease: smaller than VELmax, but not smaller than VEL8. Please be specific.**

changed in text

**Page 5, line 5: this seems to say that the total CRE is 1 W/m2 during the day. Really?**

Yes, the total cirrus CRE is about a 1 W/m² warming during daytime, where daytime is defined based on the solar zenith angle. We added a new figure (Figure 3) to the

manuscript showing annual average (a), annual day average (b), and night (c) cirrus CRE estimates as diagnosed from our model.

**Page 5, line 6-8: I'm not sure what you are saying here. cCRE in Table 3 is 4.35 W/m2. Is this 50% smaller than that in REF? Please add values for REF in Table 3.**

We are comparing the cirrus CRE for the VELmax simulation with the REF. VELmax has a cirrus CRE of about 1.1 W/m$^2$ (or, as seen from Figure 2a, the cirrus CRE is about 3.3 W/m$^2$ smaller compared to REF). REF has about 4.4 W/m$^2$, as stated in Table 3. Therefore, we reduced the cirrus CRE by about 75%, and not by the stated 50%, which was our typo that has been removed.
Rephrased in the manuscript on page 5, lines 28-30.
The VELmax simulation, which sets the cirrus IC sedimentation velocity to the unrealistically unrealistically high value (Fig. 1), shows that globally uniform cirrus cloud thinning can reduce the cirrus CRE by about 3.3 W m$^{-2}$ which is equivalent to ~75% of its full value (Table 3).

**Page 6 line 22: a lifetime of a couple of hours seems small. Are there observations to examine whether this number is reasonable?**

We estimated the cirrus lifetime by dividing the total ice water content by its source and got a result of about 2-6 hours, with the longest lifetimes at the tropical tropopause.
Yet, we note that the model-derived cirrus lifetime is highly dependent on its formulation, and may therefore well be larger.

Luo and Rossow (2004) give a lifetime of about 30 h for tropical detrainment cirrus or 12 h for tropical in-situ formed cirrus. Their lagrangian trajectory analysis is based on the ISCCP satellite dataset. Gehlot and Quaas (2012) used a similar lagrangian trajectory analysis for cirrus lifetime in ECHAM5 model and which resulted in similar tropical cirrus lifetimes. Jensen et al. (2011) on the contrary estimate the lifetime of cirrus in the tropical tropopause layer of about 12-24 hours, based on in-situ measurement data.

In summary, it is not straightforward to calculate cirrus lifetimes and the estimates have to be treated with caution.

Added in text on page 7, lines 17-20.
The effective cirrus cloud lifetime is in the range of several hours as diagnosed from our model, with values around 6 hours in the tropical tropopause region. This is shorter compared with available studies, which estimated it to 12-30 hours (Luo and Rossow, 2004, Jensen et al., 2011, Gehlot and Quaas, 2012).

**Page 7 line 1, 2: In the model description section you need to specify how cloud cover is calculated. Is it just related to relative humidity?**

Yes, it is a diagnostic cloud cover scheme by Sundqvist et al., 1989, only related to relative humidity.

We added a sentence on it in the model description, page 4, lines 6-8.
The model gridboxes are considered partially cloudy above a certain relative humidity threshold, and fully cloud covered when relative humidity reaches 100%, following Sundqvist et al., 1989.

**Page 7: lines 4 – 6: "where the IC sedimentation velocity is restored to the reference values." Doesn't this assumption essentially change the removal treatment for cirrus?**

Yes, increasing the sedimentation velocity of cirrus ice crystals changes the sinks of cirrus clouds.

**Is this a common assumption in models that alter the sedimentation velocity to simulate geoengineering?**

Yes, the sedimentation velocity increase to our knowledge is valid only for clouds at temperatures colder than -35°C, as described for example in Kravitz et al., 2015 (and is mentioned in the introduction, page 3, lines 11-16).

**Page 7, line 23: "in Vel2 all cirrus IC sediment regardless of their origin". So you are not applying the 2 m/s sedimentation only to the added INP for geoengineering?**

No, all ice crystals at T<-35°C, regardless of their origin, sediment with 2 m/s instead of their calculated sedimentation velocity.
In the seeding case, the large, geoengineered ice cystals sediment by their model-calculated sedimentation velocity. So we model either seeding by geoengineered INP particles or increase of sedimentation velocity for all ice crystals at cirrus conditions.

We added a sentence pointing that out explicitly on page 4, line 13.
The sedimentation velocity increase applies for all the cirrus ICs, regardless of their microphysical origin.

We additionally added a sentence on page 4, lines 22-25.
In our simulations we either increase the cirrus IC sedimentation velocity or seed with geoengineered INP, which sediment with the size dependent sedimentation velocities (Spichtinger et al., 2009).

**Page 8 lines 3,4: Are you surmising that these changes are responsible for increases in ICNC due to detrainment? i.e. you do not show this, so how do you know?**

Thanks for pointing that out.
The prevalent driver of the ICNC increase at mixed-phase is the way the sedimentation velocity increase simulations are set up, with the sedimentation suddenly being restored to the (smaller) standard value at temperatures warmer than -35°C. Interestingly, we observe also an increase in the global ICNC burden, as shown by Figure 4, which might be not as intuitive after showing the IWC decreases in the global average.

After re-evaluating the plots we suggest that the net ICNC increase is due to the mixed-phase cloud glaciation effect, in which the initial redistribution of IC to lower lying mixed-phase clouds leads to a positive feedback and more secondary ice crystal production in clouds at temperatures between -35°C and 0°C. Unfortunately we cannot provide a quantitative proof of this hypothesis but will explore this further in future studies.

Nevertheless, we can still attribute part of the ICNC increase to the observed increase in mid-level convection (see lower plot). In our model the ice crystal sizes of convectively detrained IC follow the parametrization by Boudala et al., 2002. Convective detrainment therefore produces large concentrations of relatively small IC.
We note that the interpretation has to be taken with care, as we show the convective type frequency only for fixed SST model setup and not the MLO setup.
In addition we also note that the convective part of the precipitation increases in the simulation VEL2 by 1.5% and in SEED1r50 by 1.2%, additionally pointing at the enhanced convective activity.

[Figure]

**Figure 1: Annually averaged occurrence frequency of deep, mid-level or deep and mid-level convection from a 5-year long fixed SST simulation. Hatching is applied for areas at 90% significance level.**

**Page 8, line 8, 9: Why does an increase in convective activity lead to a decrease in CDNC?**

The model seems to follow the response mechanism described in Rieck et al., 2012, in which the increase in convective activity decreases RH in the boundary layer. The RH is directly related to cloud cover (Sundqvist et al., 1989). A decrease in RH therefore leads to a decrease in cloud cover, which decreases also the all-sky CDNC.

Added also in text on page 9, lines 12-15.
Furthermore, in SEED1r50 an increase or intensification of convective activity , expressed by an 1.2% increase in globally averaged convective precipitation, leads to a drying of the tropical planetary boundary layer and lower troposphere and a decrease in liquid cloud cover (Fig. 4a). The cloud cover is directly related to RH (Sundqvist et al., 1989); its decrease leads to a cloud cover decrease, which decreases also the all-sky water content and CDNC (Fig. 6 a,c and Fig. 4 e).

**Page 8, line 13-15: Why is CDNC higher in mixed phase cloud?**

Added in the text, page 9 lines 21-28.
The sedimentation of ICs into mixed-phase clouds leads to the IC growth by riming of supercooled cloud droplets, also known as the seeder-feeder mechanism (Politovich and Bernstein, 1995). The seeder-feeder mechanism, reinforced by the additional growth of ice crystals at the expense of supercooled cloud droplets (Wegener-Bergeron-Findeisen process (Storelvmo and Tan, 2015)) therefore leads to a depletion of CDNC and LWC. A decrease of IC sedimenting flux from cirrus in the simulation SEED1r50 therefore leads to an increase in CDNC and LWC in mixed-phase cloud regime. This is in the tropics contrasted by the large RH decrease (Fig. 5f), leading to a decrease in cloud cover (Fig.  5a), and consequently also a decrease in all-sky CDNC in Fig. 6a.

**Page 8, line 31: "T anomalies follow cCRE anomalies" Not in general true, since you do not see warm pool anomalies.**

Thanks for pointing that out.
Indeed, that is not the case for the warm pool. We did not dig in the details of such a disagreement, as that goes beyond the scope of this publication.
Changed also in text, page 10, lines 10-11.
Yet, the temperature anomaly in the Pacific warm pool area is an exception to this general trend, which needs to be addressed in future studies.

**Page 10: line 10: Why do change in tropical convection lead to a large number of small IC and an increase in ICNC?**

This is due to the used parametrization of the formation of detrained IC, which follows Boudala et al., 2002.

Included also in the text, page 11, lines 29-33.

The surprising result can be explained by the parametrization of the size of the detrained ICs, which assumes an IC radius of about 10-20 µm (Boudala et al., 2002), distributing the detrained IWC over a large number of ICs and is part of the reason for the ICNC increase pattern in mixed-phase cloud conditions (Fig. 5w). Moreover, the freezing in mixed-phase clouds in our model occurs only rarely in-situ on dust or black carbon aerosols and is largely affected by the sedimented ICs from cirrus levels, initiating a seeder-feeder type of IC growth.

**Page 10, line 14: replace "on some climatic" with "for some climatic"**

Done

**Page 11, line 24 – 26: Zhou and Penner, JGR, 2014 show different model assumptions that are used to describe the number of homogeneous and heterogeneous particles.**

Good point, we added their assumptions on background aerosols in the text on page 13, line 14-15.
We also expect the cirrus seeding effectiveness to be dependent on the amount of background aerosol available for both homogeneous and heterogeneous freezing (Zhou and Penner, 2014).

**Page 11, line 28: what is the "inhomogeneity parameter for ice clouds"**

The model cannot resolve variability in the cloud field at scales smaller than the grid box size, where the clouds' natural variability is still large. Therefore, the use of a cloud inhomogeneity factor with values smaller than one improves the model agreement with observations of the planetary albedo and radiative fluxes (Cahalan et al., 1994). However, the uncertainty of the inhomogeneity parameter for clouds is large, and is therefore often used as a tuning parameter (Mauritsen et al., 2012).

We do not mention inhomogeneity parameter in the manuscript text any more.

**Page 11: last line: what does "to which fraction of the total cirrus CRE this radiative anomaly corresponds" mean? The fraction due to liquid or mixed phase clouds?**

Rephrased in text.

**Figure 2: Please explain why CRE is not the same as the net (all sky) results. Also, why does it switch from higher than net to lower than net?**

We explain the reasons for the disagreement between the net and cirrus CRE (and not total CRE) anomalies at the top of the atmosphere for increased sedimentation velocity setup, SEED1r50, and SEED1r50N in section 3.2.2, on page 9.

The switch between higher than net to lower than net, as observed in SEED and SEEDr50 simulations (Fig. 2 b,c) is interesting and can most likely be explained by additional cloud responses, as discussed for the VEL2, SEED1r50, and SEED1r50N simulations in the text.

**Table 1: please write out what MLO means within the table (as well as in text).**

Added to the table in addition to the main text on page 5, line 10.

**Table 5: Please add a description of how $\Delta$mpCRE, $\Delta$cCRE, $\Delta$liqCRE are computed within text.**

Included in text, in the experimental setup, on page 5, lines 3-8.

In addition to the standard model radiative fluxes, we separately diagnosed the cloud radiative effect contribution of clouds at temperatures colder than -35°C (cirrus cloud radiative effect, cCRE) with the help of the double call of the radiation routine. Similarly, we diagnosed mixed-phase cloud radiative effects (mpCRE) for all clouds at temperatures between -35°C and 0°C independent of their cloud phase, and liquid cloud radiative effects (liqCRE) for clouds at temperatures above the freezing level.

**REVIEWER 2**

This is a nice and straightforward, if not exactly Earth shattering, modeling study of geoengineering in the form of cirrus seeding. The study includes highly relevant testing of the usefulness of the approach of mimicking cirrus seeding by increasing ice crystal fall speed. The latter has been proposed as an experiment in a new set of GeoMIP simulations, so it is important to understand in what respects this is a useful proxy for explicitly simulating the shift from homogeneous to heterogeneous nucleation that (at least in theory) occurs in response to seeding with INP.

I have a few major/substantial comments, and numerous minor comments which are listed below, and which are intended to help the authors improve the readability of the manuscript. Once the major/minor comments have been addresses, I believe the manuscript will be suitable for publication in ACP.

**Major comments:**
- The authors conclude that the simulation that includes seeding at nighttime only is the most "appealing". To me, that simulation is a purely academic exercise, because in practice it would be impossible to only seed during nighttime. It would take time to build up the right seeding INP concentration, and obviously one could not make the particles magically disappear at sunrise. I understand the desire to minimize the SW radiative effect, as well as the effects of increased convection, but I don't understand why the study doesn't include simulations with only high-latitude seeding (and fall speed increases).

Seeding only high latitudes would, as opposed to the entire globe at nighttime, potentially be possible in practice, and should achieve many of the same advantages. Considering the very large particles that appear to be most favorable seeding INP in this model, it would also be advantageous to seed much smaller areas to reduce the total mass required.

We didn't want to expand the manuscript too much beyond the comparison of seeding with increased sedimentation velocity setup, which represents the core of this study. We do have a few simulations where seeding/increased sedimentation velocity has been applied only over high latitudes, with the same zenith angle dependent seeding scenarios as in Storelvmo and Herger, 2014.

Our cirrus clouds respond differently to seeding compared to what described in Storelvmo and Herger, 2014 publication. A solar zenith angle dependent seeding scenario Y1 leads to only about half of the radiative flux anomaly compared with the seeding over the whole world. The difference to Storelvmo and Herger's publication probably lies in the radiative effects of our cirrus clouds, which show a peak over the tropics and a large proportion of tropical cirrus clouds being formed by homogeneous freezing, as shown by Gasparini et al., 2016.

Similarly, increasing sedimentation velocity using the solar zenith dependent scenario Y1 also leads to only about half as large radiative anomaly effects compared to the globally uniform sedimentation velocity increase simulation VEL2.

Added also in the text on page 6/7, lines 31-34/1-2:

Interestingly, our cirrus clouds respond differently to seeding compared to what has been described in Storelvmo and Herger, 2014. A solar zenith angle dependent seeding scenario which seeds about 40% of the earth's surface leads, unlike the cited study, to only about half of the radiative flux anomaly compared with the globally uniform seeding strategy SEED1r50. The difference probably originates from the zonally different radiative effects of cirrus clouds in ECHAM-HAM, which show a peak over the tropics (Fig. 3) and, differently from Storelvmo and Herger, 2014, a large proportion of tropical cirrus clouds is formed by homogeneous freezing (Gasparini et al., 2016).

We will revisit limited seeding scenarios in a future publication.

The night seeding scenario is indeed less realistic compared with other experiments (added a comment on that in the manuscript, page 13, line 6: "…not considering its unlikely technical implementation,…". But we would argue that none of our experiments is much more than an academic exercise. We seed at every model timestep at all locations where the cirrus formation scheme is called (i.e. at T<-35°C, in conditions of updraft, at $RH_{ice} > 100\%$). That indeed is not a very plausible scenario either.

Yet, in defence of the night seeding scenario, we want to point out that such large INP forming large ICs can sediment out of the atmosphere (or at least out of upper troposphere) in only a few hours time. Large particle seeding could in this case instead of building up a permanent aerosol layer be used to target specific regions, in which one would forecast conditions favourable for homogeneous ice nucleation.

**Considering the very large particles that appear to be most favorable seeding INP in this model, it would also be advantageous to seed much smaller areas to reduce the total mass required.**

This is a good point and we also started to explore it, by limiting seeding for example to areas of prevalent homogeneous freezing, globally non-uniform seeding INP concentrations, etc. It will be subject to a future publication.

Again, we avoided including that in the manuscript to prevent it from becoming too large and to keep the focus on the main scientific question of this manuscript – the comparison of sedimentation velocity increase to cirrus seeding.

We added a sentence to conclusions, page 13, lines 8-10.

We note that a seeding strategy limited to areas of highest seeding effectiveness (where the cCRE anomalies after seeding are the largest as shown by Fig. 7a), might significantly decrease the mass of seeded material while exerting a roughly similar climatic forcing.

**- The simulations with increased fall speed are, while in some respects useful, deeply unphysical. I missed a discussion of this in the manuscript. Specifically, I have problems with the sudden drop in fall speed that all ice crystals will experience as soon as they fall through the -35 degree isotherm. This would naturally lead to an accumulation of ice at mixed-phase levels, which is exactly what can be seen in Fig. 4. The authors seem to attribute this to mid-level convection. This is one important reason why increased fall speed is an imperfect proxy for actually simulating cirrus cloud microphysics.**

We totally agree on the sudden drop in fall speed as the main reason for the increase in IC number concentration and IWC in the model. We revised the text to make it clearer.

In addition, we still argue that increases in convection (or more specifically, mid-level convection) do contribute towards part of the ICNC increase and RH signal. Mid-level convection in our model has an upper limit of 400 hPa, with the detrainment occurring in the 2 levels above it, that is between 400 and 300 hPa. This is exactly the level where the RH increases, in particularly in the VEL2 and VEL2all simulations. Moreover, an increase in convective precipitation in VEL2 by about 1.5% in global average additionally indicates a likely convective strength increase.

Please see also changes in text, which better point out the problem of the drop in IC velocity, for instance in conclusions, page 12, lines 9-11.
However, in VEL2 the IC sedimentation velocity is abruptly set back to the standard one computed by the model at temperatures warmer than -35°C, leading to a redistribution of ICs and IWC from cirrus to underling mixed phase clouds, which is not observed in SEED1r50.

**Minor comments:**
**- Make sure you're consistent in your use of the abbreviations IC and INP - they are used to represent both the plural and singular forms of the nouns.**

Thanks, we corrected it and tried to be more consistent in its use now.

**- Page 4, Line 20: "precipiation"–>"precipitation"**
**- Page 5, "for"–>"by"**
**- Page 5, Line 31-31: inappropriate referencing**
**- Page 7, Line 11: suggest replacing "affects also part of" with "to some extent also affects"**
**- Page 7, Line 19: suggest replacing "temperature decrease anomaly" with**

**"negative temperature anomaly"**
**- Page 11, Line 15: "twice larger"–> "twice as large"**

Changed in text

**- Figure 4: Why are ice crystal numbers and sizes not included in this figure? They seem like very important microphysical variables to include.**

We did not include IC number concentration and size to prevent increasing the complexity of the figure. The message however does not change when adding the additional plots. Yet, as the change in ICNC and IC radius do matter when trying to interpret the climatic responses to seeding, we decided to add them nevertheless to the manuscript.

Please note also changes to the text on pages 7, 10, 11 related to the inclusion of ICNC and IC radius in Figure 4 (now Figure 5).

**- Figure 5: Are these in-cloud or grid-box values?**

all-sky values, added to the figure caption

**REVIEWER 3**

This is a manuscript on evaluating a simplified geo-engineering simulation strategy against the more comprehensive microphysical model treatment. The authors performed both fixed-SST and mixed-layer ocean simulations and investigated the changes in cloud properties for both ice and liquid phases, and the associated changes in cloud radiative forcing, surface temperature, and precipitation, due to using different geo-engineering model setups. The manuscript is clearly written and well organized. The designed experiments and analysis are comprehensive. The conditionally sampled/calculated CRE changes are interesting and informative. I only have several questions that need clarification and some minor suggestions for better readability. Below please find my specific comments:

1. Page 2, Line 20: Cirrus seeding affects the sedimentation of ice crystals from cirrus clouds to mixed phase/warm clouds, the cloud glaciation, and therefore the ice supersaturation. Maybe change it to "cannot directly influence"?

Thanks, changed in the text.

2. Page 3, Line 12: What is the model time step? This is related to the question below about the numerical treatment of ice sedimentation.

It's 6 minutes.

3. Page 3, Line 26-27: If the model time step is about 20min, an ice crystal with a sedimentation velocity larger than 1m/s will fall through a 800m thick model layer (800m/1200s=0.67m/s). How does the model treat this (violation of CFL condition)?

We use a time step of 6 min, i.e. 360 sec so CFL is not violated.
Added also to the text, page 3, line 25.
...and a model timestep of 6 minutes.

4. Page 4, Line 5-6: What is the initial size of the nucleated ice particles? If not the size of the seeded INP, is it determined by the parameterization or by explicit microphysical calculation?

The aerosol both nucleates an ice crystal as well as experiences the initial growth by deposition within the same model timestep. Therefore the ice crystals rapidly grow beyond their initial sizes.
The size of the newly nucleated ICs is limited by the INP size and/or a minimum allowable IC size of 1 μm (Kärcher and Lohmann 2002, Kärcher and Lohmann 2003, Kärcher et al., 2006).

**5. Page 4, Line 12: Does the seeded INP immediately freeze at RHi=105% when T<-35C? How do you consider the competition between the homogeneous freezing of solution droplets, heterogeneous freezing of natural dust, and heterogeneous freezing of the seeded INP?**

It can nucleate if the $RH_{ice}$ exceeds 105% at the moment the cirrus scheme is called by the model.
After the cirrus scheme is called (always when $RH_{ice}$ > 100% and in presence of an updraft) the scheme verifies whether cirrus conditions were met, in the following order (from the most effective INP onwards):
  1.) seeding by deposition
  2.) heterogeneous nucleation – deposition
  3.) heterogeneous nucleation – immersion
  4.) homogeneous freezing

In case of pre-existing ice crystals, the cirrus scheme first computes the depositional growth of water vapor onto these pre-existing ice crystals, decreasing the $RH_{ice}$.
If the $RH_{ice}$ is still high enough, the freezing proceeds as mentioned above, starting from the most effective freezing mechanism (in our case seeding). If no seeding INP are present, the scheme tries to freeze ice crystals heterogeneously, and finally, homogeneously. After each freezing (and initial growth) step, the $RH_{ice}$ is decreased accordingly.

Technically, the $RH_{ice}$ is represented by the so-called 'fictitious updraft'. More information about its details can be found in Kuebbeler et al., 2014 and Kärcher et al., 2006.

**6. Page 4, Line 13-14: Do you avoid the INP seeding in anvil clouds? Or in-situ ice nucleation doesn't happen in anvil clouds?**

We seed in the described setup also anvil clouds. Yet, the in-situ nucleation occurs rarely under such conditions.
Added in the text, page 4, lines 33-34.
It is important to note that we only modify in situ formed cirrus and not the convective anvil clouds as in situ deposition nucleation does not occur in anvils.

**7. Page 5, Line 24-25: I don't quite understand this. Do you mean with the same amount of ice crystal mass?**

The freezing probability increases with the surface area of particle, which quadratically depends on its radius. The sedimentation velocity of INP increases quadratically with the radius too. The two effects cancel out each other.

Rephrased in text on page 6, line 15-18.
On the other hand, the probability P of one INP to freeze in a given time as described by the classical nucleation theory depends on the INP's surface area, which

increases quadratically with particle size. Quadratic fallspeed velocity and freezing probability increases cancel out each other leading to no change in the concentration of ICs formed on geoengineered INPs when they increase in size.

**8. Page 8, Line 27: The signal over topography seems pretty strong. Does the model consider the impact of orographic waves on ice nucleation?**

Yes – we follow the Joos et al., 2008 formulation, as mentioned in the referenced studies with the current cirrus scheme (Gasparini and Lohmann, 2016 and Kuebbeler et al., 2014). We added the following 2 sentences in the manuscript on pages 3/4, lines 34/1-2.

The formulation of vertical velocity used for cirrus formation considers the large-scale velocity field and a sub-grid scale contribution derived from the turbulent kinetic energy. The latter is replaced by a gravity wave parametrization by Joos et al., 2008 over mountain regions.

**9. Page 9, Line 28-29: This result is very interesting, but it's very likely model-dependent. Do you have extinction output as well? Would be nice to show the forcing efficiencies (&CF/&EXT) in table 3 as well.**

Unfortunately we do not have the extinction or cloud optical depth output for the performed analysis. Yet, we expect the efficiency to be the largest at the coldest cirrus clouds, despite its absolute forcing being very small.

**10. Conclusion: Many points are made in the conclusion part and to me, they are a little bit scattered (very useful information though). I would recommend the authors to make it more compact and concise.**

**Maybe in the order of 1) statement of general findings;**
**2) differences in microphysical responses;**
**3) differences in CREs;**
**4) differences in temperature and precipitation response; . . .?**

Thanks for the suggestion, we tried to modify the conclusions according to your suggestion. The reviewed conclusions section has now the following order:

-general findings (with microphysical responses, 1 paragraph)
-differences in CRE (2 paragraphs)
-differences in temperature and precipitation response (2 paragraphs)
-general conclusions (last 2 paragraphs)

Kärcher and Lohmann, 2002: A parameterization of cirrus cloud formation: Homogeneous freezing including effects of aerosol size

Kärcher and Lohmann, 2003: A parameterization of cirrus cloud formation: Heterogeneous freezing

[revised manuscript text omitted]